## SOFTWARE

# SuperCellCyto: enabling efficient analysis of large scale cytometry datasets

Givanna H. Putri[1*], George Howitt[2], Felix Marsh-Wakefield[3], Thomas M. Ashhurst[4] and Belinda Phipson[1*]

*Correspondence:
putri.g@wehi.edu.au;
phipson.b@wehi.edu.au

[1] The Walter and Eliza Hall Institute of Medical Research and The Department of Medical Biology, The University of Melbourne, Parkville, VIC, Australia
[2] Peter MacCallum Cancer Centre and The Sir Peter MacCallum, Department of Oncology, The University of Melbourne, Parkville, VIC, Australia
[3] Centenary Institute of Cancer Medicine and Cell Biology, The University of Sydney, Sydney, NSW, Australia
[4] Sydney Cytometry Core Research Facility and School of Medical Sciences, The University of Sydney, Sydney, NSW, Australia

## Abstract

Advancements in cytometry technologies have enabled quantification of up to 50 proteins across millions of cells at single cell resolution. Analysis of cytometry data routinely involves tasks such as data integration, clustering, and dimensionality reduction. While numerous tools exist, many require extensive run times when processing large cytometry data containing millions of cells. Existing solutions, such as random subsampling, are inadequate as they risk excluding rare cell subsets. To address this, we propose Super-CellCyto, an R package that builds on the SuperCell tool which groups highly similar cells into supercells. SuperCellCyto is available on GitHub (https://github.com/phips onlab/SuperCellCyto) and Zenodo (https://doi.org/10.5281/zenodo.10521294).

**Keywords:** Cytometry, Cytof, Dimensionality reduction, Computational analysis, CITEseq, Supercell, Batch correction, Clustering, Data compression, Bioinformatics

## Background

Cytometry technologies, such as flow cytometry and mass cytometry, have revolutionised the field of cellular biology by enabling the high-throughput quantification of physical and biochemical characteristics of millions of cells at the single cell level. These technologies, which can measure cellular dimensions, granularity, and the expression of over 40 proteins (markers), have found widespread application across diverse biological and medical research domains. For instance, they have been used to elucidate immune responses to infectious diseases [1], to investigate B cell behaviour in Multiple Sclerosis patients [2], and to study the human hematopoietic system [3].

Traditional cytometry data analysis typically employs manual gating, a process which delineates cell types by iteratively refining polygons on a series of two-dimensional scatter plots, with each plot illustrating the expression of two markers. However, this approach is intractable for identifying a vast variety of cell types due to the numerous combinations of two markers that must be examined [4]. To address this, a plethora of computational methods have been developed. These include methods for data pre-processing (such as CATALYST [5]), clustering (such as FlowSOM [6], Phenograph [7],

X-shift [8]), dimensionality reduction (such as UMAP [9] and FITsne [10]), and differential expression or abundance analysis (such as Diffcyt [11] and Citrus [12]). Additionally, toolkit-like software libraries such as Spectre [4], CytofKit [13], and CyTOF workflow [14] provide function interfaces (wrappers) for a range of computational methods, thereby facilitating the creation of analysis workflows. Concurrently, the single cell RNA sequencing (scRNAseq) field has seen an explosion in bioinformatics methods development, with more than 1,500 tools currently available [15], including Seurat [16–19], Bluster [20], Batchelor [21], and Speckle [22]. There are many computational analysis tasks that are shared between cytometry and scRNAseq data analysis, e.g., clustering and batch correction. However, methods to analyse cytometry and scRNAseq data have been mostly developed independent of one another. We believe there is an opportunity to take advantage of the many scRNAseq tools for analysing cytometry data.

Despite the robustness and diversity of existing computational methods, many face challenges in terms of scalability when processing large cytometry datasets [4]. This issue is particularly pronounced when using methods primarily developed for scRNAseq data [23]. Given that scRNAseq data typically encompassess fewer cells but a significantly higher number of features compared to cytometry data, these methods may require an excessive runtime when applied on cytometry data, making them impractical for efficient data analysis. Importantly, this issue is not confined to scRNAseq methods alone, and is also evident in some cytometry-specific methods [24]. Furthermore, many computational methods depend on parameters that significantly influence their performance, often necessitating time-consuming parameter tuning for optimal results. Given the continuous expansion of cytometry data, which now routinely includes millions of cells, the demand for more efficient methods is increasingly urgent.

Existing strategies to address these challenges include data reduction through random subsampling, which, although expediting analysis, may result in the loss of representation for rare cell populations. Alternative solutions such as method re-implementation or utilisation of high-performance computing (HPC) platforms may not be feasible due to the need for specialised expertise and/or limited accessibility to HPC resources.

A notable alternative is the HSNE algorithm, as implemented in the Cytosplore software [25, 26]. This method hierarchically aggregates cells, allowing for their visualisation and interactive exploration. Cytosplore allows users to identify cell types by either manually selecting and annotating them, or by clustering them using the Gaussian Mean Shift algorithm at various levels of aggregation. However, while Cytosplore excels in interactive data exploration, its utility can be limited for users who prefer to use open-source computational tools or packages such as Spectre [4] for more complex, customised analyses. In this study, we present a strategy to mitigate these challenges by developing SuperCellCyto, an adaptation of the SuperCell R package [27]. Initially developed for scRNAseq data, the SuperCell method aggregates cells with similar transcriptomic profiles into "supercells" (also known as "metacells" in the scRNAseq literature [28–30]). This aggregation effectively reduces the dataset size by 10 to 50 times, while preserving biological diversity, thus alleviating the computational demands of downstream analysis.

SuperCellCyto complements software like Cytosplore. It specifically caters to users who prefer to conduct sophisticated tailored analyses using open-source tools or packages, addressing a niche that traditional Graphical User Interface based softwares may

not fully cover. It is a significant enhancement of the SuperCell package. Specifically, we have implemented within-sample supercell creation, preventing supercells from containing cells across multiple samples. Additionally, we have introduced parallel processing capabilities using a careful load balancing strategy, allowing the simultaneous creation of supercells across multiple samples, which significantly speeds up computational time.

We highlight supercells' capacity to retain biological heterogeneity by aggregating single cells in previously annotated cytometry data into supercells and verify that each supercell predominantly comprises unique cell types. Furthermore, we showcase the viability of analysing supercells as a surrogate to single cells by conducting a series of downstream analyses on supercells derived from six publicly available cytometry datasets. These analyses encompass cell type identification, batch effect correction, differential expression and abundance analysis. Finally, we demonstrate the ability to annotate supercell-level cytometry data using a multiomics dataset that measures RNA and protein expression on the same cells (Cellular Indexing of Transcriptomes and Epitopes (CITEseq) [31]) as a reference. This allows us to exploit the rich cell type annotation derived from deep transcriptome sequencing. Altogether, our findings affirm that Super-CellCyto efficiently reduces dataset size, vastly reduces the computational burden of analysing large cytometry datasets, and maintains the integrity of downstream analyses.

The SuperCellCyto R package is publicly available on GitHub (https://github.com/phipsonlab/SuperCellCyto) [32] and Zenodo (https://zenodo.org/records/10521294) [33], along with extensive vignettes (https://phipsonlab.github.io/SuperCellCyto/) [34], providing a valuable resource for the research community to incorporate SuperCellCyto into their analysis pipelines more efficiently and effectively. Additionally, the complete analysis workflow associated with the results presented below can be accessed online on https://github.com/phipsonlab/SuperCellCyto-analysis [35].

## Results

### Concurrent generation of supercells across multiple samples using SuperCellCyto

The SuperCellCyto R package is an extension of the SuperCell package adapted specifically for cytometry data. SuperCellCyto offers a practical approach to reduce the size of large cytometry data by grouping cells with similar marker expression into supercells. Figure 1A depicts the schematic overview of the generation of supercells using the SuperCellCyto R package. The process begins with performing a Principal Component Analysis (PCA) on the cytometry data to capture the main sources of variation. The number of principal components (PCs) is adjustable, with a default setting of 10, but can be increased up to the number of markers in the data. For datasets with fewer than 10 markers, the number of PCs is set to the number of markers in the data. Using the PCs, a k-nearest-neighbour (kNN) graph is constructed, with each node representing a single cell. A walktrap algorithm [36] is then applied to identify densely interconnected subgraphs or communities. This step involves performing a series of four-step random walks from each node, where a single step represents a transition from one node to another. The destination node for each step is selected randomly, and the probability of a random walk starting and ending at a given pair of nodes is used to determine their proximity. After computing these distances, nodes are iteratively merged, starting with each node in its own community, until a single

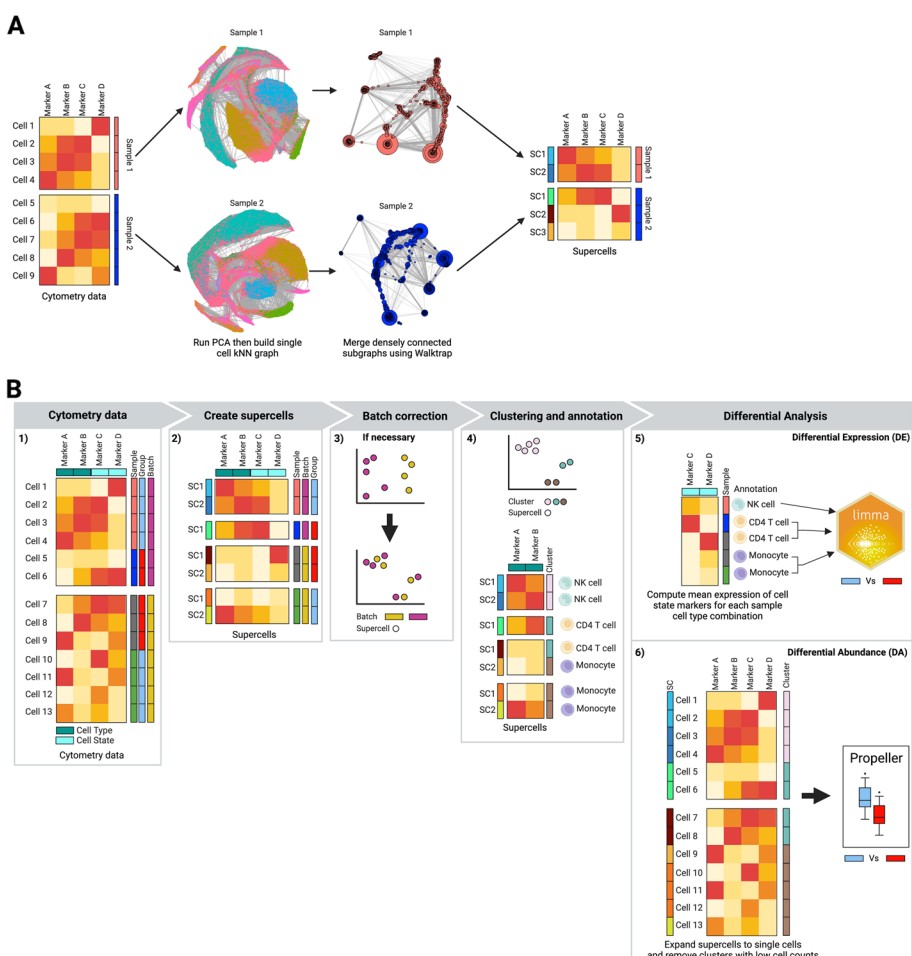

**Fig. 1** The SuperCellCyto framework. **A** Schematic overview of the SuperCellCyto R package for generating supercells from cytometry data. **B** Proposed workflow for an end-to-end cytometry data analysis using supercells as the fundamental unit of analysis. The workflow begins with the generation of supercells using SuperCellCyto from cleaned and transformed cytometry data. If necessary, batch correction is then performed at the supercell level. Supercells are then clustered and annotated based on the cell type they represent. Differential expression analysis can be performed using existing tools such as Limma following aggregation of supercells at the sample and cell type level. Differential abundance analysis is performed after expanding supercells back to single cells using existing tools such as Propeller

community is formed, creating a dendrogram-like structure. This dendrogram is then cut at a specific level to yield the desired number of communities, referred to as supercells.

The number of supercells generated for a given dataset is controlled by the parameter gamma, which governs the granularity of the supercells and determines at which level the dendrogram is cut. Gamma is defined as the ratio of the number of cells to the number of supercells. Choosing the appropriate gamma value involves balancing the trade-off between the data compression level and the probability of supercells encompassing diverse cell types. A larger gamma value produces fewer supercells (higher compression level), but each supercell is more likely to contain more heterogeneous cell types. Conversely, a smaller gamma value yields more supercells (lower

compression level), but each supercell is less likely to contain multiple different cell types. It is important to note that each supercell may contain a different number of cells, centred around the value of gamma. We found the default value of 20 to work well for most datasets. Critically, we have implemented within SuperCellCyto the ability to adjust the number of supercells generated for a dataset without having to regenerate the kNN graph or re-running the walktrap algorithm. This feature allows users to rapidly fine tune the data compression level as required.

SuperCellCyto is designed to process each sample independently, and we have introduced the ability to process multiple samples concurrently. This is facilitated through the use of the BiocParallel R package [37] and a custom load balancing strategy. Samples are first ordered based on their cell count, and parallel workers are then tasked to process these samples in descending order, starting with the sample that contains the most number of cells. This strategy allows workers processing smaller samples to be assigned additional samples, while those handling larger samples can concentrate on their tasks without being overburdened. This approach maximises throughput and minimises idle time, thereby enhancing overall efficiency.

The SuperCellCyto R package takes as an input a transformed (using ArcSinh or Logicle [38] transformation) cytometry dataset formatted as an R data.table object [39] where each row represents an individual cell and each column denotes a marker. For each supercell and marker, SuperCellCyto calculates the marker expression of supercells by aggregating the marker expression of all cells within the supercell using either the mean or median as determined by the user. These aggregated marker expressions are outputted as an R data.table object.

Additionally, SuperCellCyto generates a cell-supercell map, also in R data.table format. This map lists each cell's supercell ID, enabling users to expand individual supercells without extra computational effort. This map is particularly useful when used in conjunction with the marker expressions of individual cells, as it allows users to access marker expressions of all the cells in a specific supercell.

Importantly, when the number of supercells is adjusted, both the supercell marker expression and the cell-supercell map are updated accordingly.

Figure 1B illustrates our proposed workflow for an end-to-end cytometry data analysis, using supercells as the fundamental unit of analysis. The first step is to pre-process the cytometry data to exclude doublets and dead cells, transform the data using either ArcSinh transformation or Logicle [38] transformation, and create supercells from the cleaned transformed single cell level cytometry data. Once supercells have been created, almost all subsequent analysis tasks can be performed at the supercell level, including batch correction (if necessary), clustering and cell type annotation, and differential expression analysis. For differential abundance analysis, we strongly recommend expanding the annotated supercells to the single cell level and calculating cell type proportions from the single cell level data. Additionally, we also recommend discarding underrepresented clusters, that is, clusters that only capture a small number of cells from each sample. For differential expression and abundance analyses, existing R packages, including those from the scRNAseq field such as Limma [40], EdgeR [41–43], or Propeller [22], can be used.

**Supercells preserve biological heterogeneity and facilitate efficient cell type identification**

We assessed whether supercells could preserve the biological diversity inherent in a cytometry dataset, and whether the clustering of supercells could expedite the process of cell type identification without compromising accuracy. We generated supercells for two publicly available cytometry datasets, Levine_32dim [7] and Samusik_all [8] (Additional file 1: Table S1) using a range of gamma values (Additional file 1: Table S2 and Additional file 1: Fig. S1). For gamma set to 20 (the default), the Samusik_all dataset was reduced from 841,644 single cells to 42,082 supercells, an approximately 20 fold reduction. Importantly, each supercell may capture a different number of cells, centred around the gamma value. The distribution of the number of cells captured in the supercells is available in the Additional file 1: Fig. S2.

Thereafter, we clustered the supercells using FlowSOM [6], a popular clustering algorithm for cytometry data, and Louvain [44], a popular clustering algorithm for analysing scRNAseq data. For both algorithms, we broadly explored their parameter space, namely the grid size and the number of metaclusters for FlowSOM, and the parameter $k$ for Louvain (see Additional file 1: Table S3 for the list of values explored). Using the cell type annotation acquired through a manual gating process performed by the authors of the datasets as the ground truth, we evaluated the quality of the supercells by using two metrics; purity and Adjusted Rand Index (ARI). Purity quantifies the homogeneity of cell types within each supercell by measuring the proportion of the most dominant cell type within a given supercell. ARI measures the similarity between the clustering of supercells and the manually gated cell type annotation. Additionally, for ARI, we also measured the concordance between the clustering of supercells and the clustering of single cells (see Materials and Methods section for more details).

Figure 2A illustrates the distribution of supercells' purity scores across all gamma values. For both datasets, we observed very high mean purity scores across all gamma values (mean purity > 0.9, Fig. 2A), with the vast proportion of supercells attaining a purity score of 1 (Additional file 1: Table S4). We compared the purity of randomly assigned cell groups with that of supercells (Additional file 1: Fig. S3). This comparison demonstrated a vastly superior purity score achieved by SuperCellCyto, which consistently shows an average purity > 0.9. In stark contrast, random grouping typically results in a much lower purity, mostly around 0.3 and 0.4.

Examining the cell types captured in each supercell, we found that the majority of supercells contained exclusively one cell type (Additional file 1: Fig. S4). While there exist instances of supercells capturing two or more cell types, they were markedly fewer. This indicates that most supercells are composed of either exclusively or predominantly a single cell type. As the gamma value increases, we observe a slight decline in both the mean purity score and the proportion of supercells obtaining a purity score of 1 (Fig. 2A and Additional file 1: Table S4) and increase in the number of supercells capturing more than one cell type (Additional file 1: Fig. S4), consistent with the fact that larger gamma values result in fewer supercells and thus higher likelihood of each supercell capturing multiple cell types. Lastly, we compared the distribution of marker expression between the supercells and single cells, and found most of them to be almost identical (Additional file 1: Fig. S5).

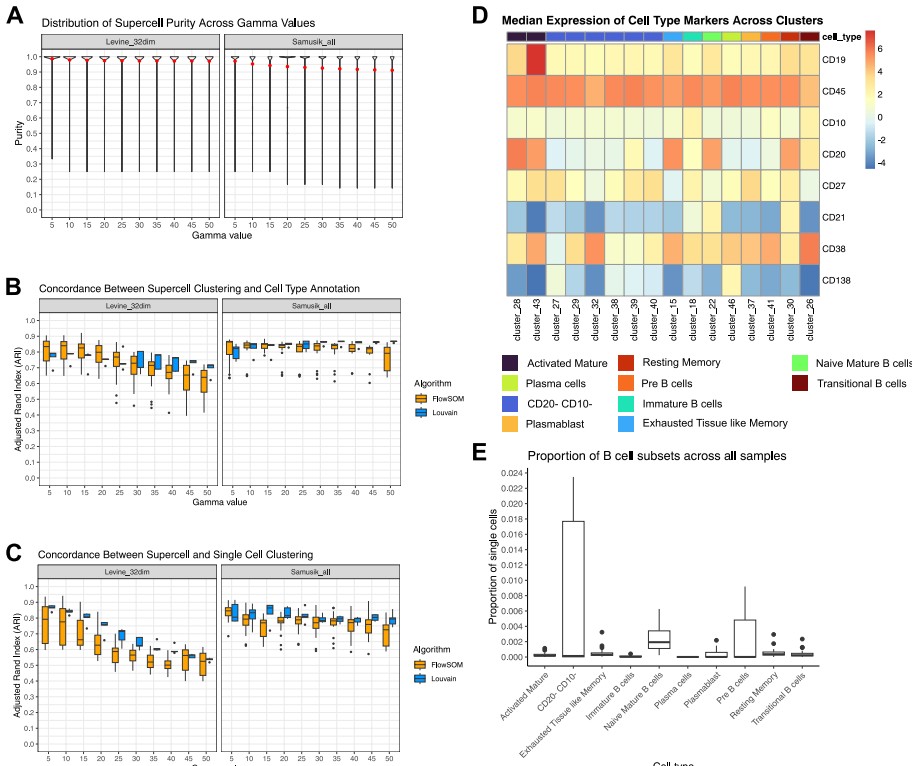

**Fig. 2** SuperCellCyto preserves biological heterogeneity. **A** Distribution of supercell purity for Levine_32dim and Samusik_all datasets across different gamma values. Red dot denotes mean purity of the supercells. **B** Adjusted Rand Index (ARI) illustrating the agreement between supercell clustering and manually gated cell type annotation for Levine_32dim and Samusik_all datasets. **C** ARI comparison between supercell and single-cell clustering. **D** Identification of rare B cell subsets through clustering of supercells. Median expression of cell type markers across supercells for each cluster and B cell subsets for the Oetjen_bcell dataset. **E** The abundance of B cell subsets across all samples as identified by clustering supercells generated for the Oetjen_bcell dataset. Cluster 46 represents the extremely rare Plasma cells which make up only 0.002% of the cells. Only clusters annotated with B cell subsets are shown

Upon examining the ARI computed between the supercell clustering and the cell type annotation (Fig. 2B), as well as between the clustering of supercells and single cells (Fig. 2C), we observed high scores across all datasets, clustering algorithms, and gamma values. Similar to the trend observed for purity scores, the ARI scores also exhibited a slight decline as the gamma value increased, due to each supercell encompassing a more diverse set of cell types. We noted a larger variation in ARI scores obtained for FlowSOM clustering results compared to those obtained for Louvain clustering results, potentially due to the broader range of parameter settings explored for FlowSOM (Additional file 1: Table S3).

These results collectively demonstrate that supercells can vastly improve clustering efficiency while effectively preserving the biological heterogeneity within a dataset, thereby enabling accurate clustering and identification of cell types.

### Identifying rare B cells subsets by clustering supercells

Downstream analysis of cytometry data routinely involves clustering cells and subsequently manually annotating them based on their marker expression to determine the

cell types they represent. While we have demonstrated that SuperCellCyto can maintain fidelity of cell types compared to cell type annotation obtained through manual gating, we next sought to verify whether we can faithfully replicate the traditional clustering and annotation process, and identify rare cell populations at the supercell level. We applied SuperCellCyto and Louvain clustering to a large flow cytometry dataset profiling more than 8 million B cells in healthy human bone marrow samples (Oetjen_bcells data [45], Additional file 1: Table S1).

Our analysis workflow consists of steps 1, 2, and 4 in Fig. 1B. First, we ran SuperCell-Cyto with gamma set to 20 to reduce the dataset to 415,711 supercells. We next clustered the supercells using the Louvain clustering algorithm. Based on the clustering results and known marker expression for B cells (Fig. 2D and Additional file 1: Note S1), we successfully identified all the B cell subsets present in the dataset, including the extremely rare plasma cells of which there are only 162 cells (0.002%) present at the single cell level (Fig. 2E and Additional file 1: Fig. S6A). To further validate the cell type annotation conducted at the supercell level, for each cell type, we expanded the supercells into individual cells and examined their marker expression profiles. We found them to be consistent with the manual gating scheme previously used by Oetjen et al. to identify the B cell subsets (Additional file 1: Fig. S6B and Note S1).

To demonstrate the effectiveness of SuperCellCyto, we conducted a comparison using the same dataset. From the same dataset (Oetjen_bcells data), we randomly subsampled 415,711 cells, which matches the number of supercells generated by SuperCellCyto, and then clustered them using Louvain clustering. With this subsampled data, we identified 7 out of the 10 available B cell subsets, with the loss of Activated Mature B cells, Plasma cells and Plasmablast (Additional file 1: Fig. S7). In contrast, using SuperCellCyto, we successfully identified all 10 B cell subsets, including the rare plasma cells. This result clearly demonstrates SuperCellCyto's superior ability in preserving biological heterogeneity within the data.

### Mitigating batch effects in the integration of multi-batch cytometry data at the supercell level

Cytometry experiments often generate datasets comprising millions of cells across several batches. This can introduce technical variation, known as batch effects, between samples from different batches. Batch effects stem from differences in experimental conditions and/or instruments [4]. Before proceeding to downstream analyses like clustering or differential expression or abundance analyses, it is imperative to rectify these batch effects using batch correction methods such as CytofRUV [46] or cyCombine [47] (for more details, see Materials and Methods section).

To demonstrate the feasibility of correcting batch effects at the supercell level, we applied CytofRUV and cyCombine to a large mass cytometry dataset profiling the peripheral blood of healthy controls (HC) and Chronic Lymphocytic Leukaemia patients (CLL) (Trussart_cytofruv data [46], Additional file 1: Table S1). This dataset consists of 8,589,739 cells and 12 paired samples profiled across two batches (each batch profiles one of the paired samples), yielding a total of 24 samples (Fig. 3A). Our analysis workflow consists of steps 1–3 in Fig. 1B. We applied SuperCellCyto with

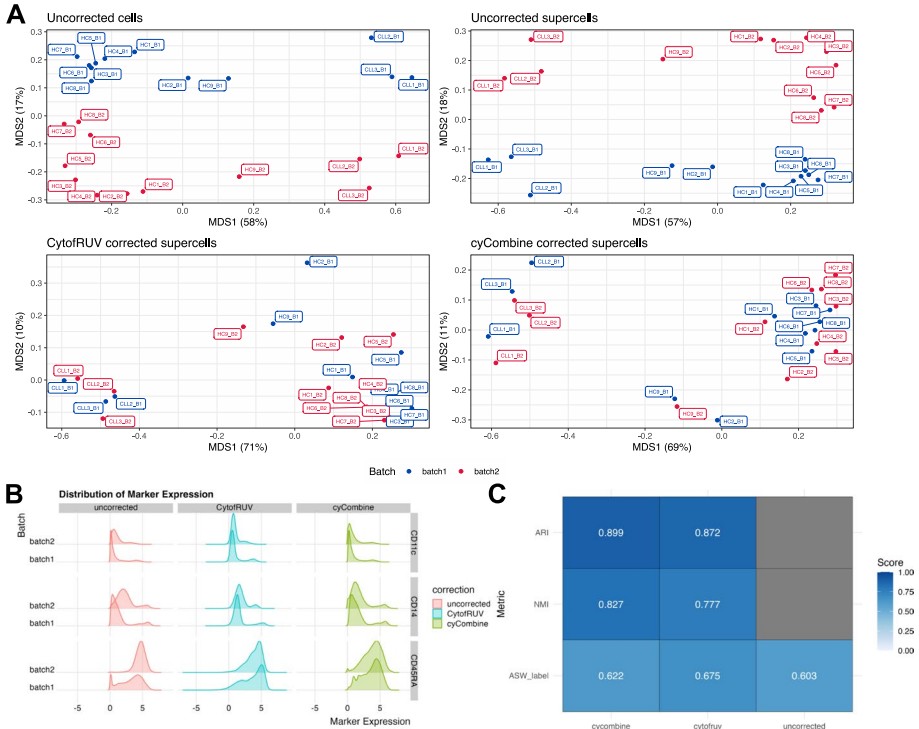

**Fig. 3** Application of batch correction techniques at the supercell level. **A** A Multidimensional Scaling (MDS) plot showcasing the variation in uncorrected single cells, supercells, and supercells that have been batch corrected using the CytofRUV and cyCombine algorithms for the Trussart_cytofruv dataset. Each point represents a sample. **B** Distribution of CD14, CD45RA, and CD11c expression for uncorrected, CytofRUV-, and cyCombine-corrected supercells for Trussart_cytofRUV dataset. **C** Scores for metrics used to assess the preservation of biological signals of cyCombine and CytofRUV. ARI and NMI were only calculated for corrected supercells, as these metrics require a comparison with respect to the uncorrected supercells

gamma set to 20 to reduce the data to 429,488 supercells. Importantly, as per SuperCellCyto's default mechanism, the supercells for each sample were generated independent of other samples. This ensures that there is no mixing of cells from different samples or batches within any supercell. Following the creation of supercells, we employed CytofRUV and cyCombine for batch correction.

The effectiveness of the batch correction methods was assessed through Multidimensional Scaling (MDS) analysis and comparison of the Earth Mover Distance (EMD) metric for each marker across the two batches. In addition, we also used 3 different metrics to assess the preservation of biological signals post batch correction—Adjusted Rand Index (ARI), Normalised Mutual Information (NMI), and Average Silhouette Width for labels (ASW_label). These metrics are taken from the scib package [48], renowned for benchmarking batch integration tools for scRNAseq data. Both ARI and NMI evaluate the consistency of clustering results pre- and post-batch correction while ASW_label quantifies the compactness and separation of clusters. For these metrics, the cluster labels were obtained by clustering the supercells before and after batch correction using FlowSOM [6]. All metrics were by default scaled to yield scores between 0 and 1 by the scib package, where 0 denotes poor while 1 denotes excellent performance.

Figure 3A presents the MDS plots at the sample level for uncorrected single cells, uncorrected supercells, and CytofRUV- and cyCombine-corrected supercells. In both uncorrected single cells and supercells, dimension 1 distinguishes CLL samples from HC samples, while dimension 2 separates the batches, indicating the presence of strong batch effects even at the supercell level. However, in the batch-corrected supercells, while dimension 1 continues to differentiate the CLL samples from the HC samples, dimension 2 no longer separates the samples based on their batch. This observation is further supported by the UMAP plots of uncorrected and corrected supercells, which demonstrate reduced batch-based separation following correction (Additional file 1: Fig. S8).

We compared the marker expression distribution between the two batches for uncorrected, CytofRUV-corrected, and cyCombine-corrected supercells, and found them to be more similar to one another following batch effect correction (Additional file 1: Fig. S9). This was particularly evident for CD14, CD45RA and CD11c markers (Fig. 3B).

The EMD calculated for the uncorrected and corrected supercells are shown in Additional file 1: Fig. S10. Generally, we observed a reduction in EMD score for both Cytof-RUV- and cyCombine-corrected supercells, indicating effective batch effect correction. Alongside the reductions in EMD scores, the visual examination of marker expression in Fig. 3B and Additional file 1: Fig. S9 indicates that the marker expression distribution post batch correction is largely preserved and not being unduly compressed (over-corrected).

Figure 3C illustrates the results for the 3 metrics employed to evaluate the preservation of biological signals following batch correction. For both cyCombine and CytofRUV corrected supercells, we observed high scores for ARI and and NMI, indicative of strong concordance in the clusterings of uncorrected and batch corrected supercells. Furthermore, there was an increase in the ASW_label score post batch correction, denoting denser and better separated clusters. Altogether, these metrics further suggest an effective preservation of biological signals post batch correction.

In summary, our analysis demonstrates that while batch effects are present at both the single cell and supercell level, they can be effectively corrected using batch effect correction methods such as CytofRUV and cyCombine following summarisation of single cells to supercells with SuperCellCyto.

### Recovery of differentially expressed cell state markers across stimulated and unstimulated human peripheral blood cells

For cytometry data, a typical downstream analysis following clustering and cell type annotation is identification of cell state markers that are differentially expressed across different experimental groups or treatments. In this analysis, we assessed whether a differential expression analysis performed at the supercell level can recapitulate previously published findings obtained by performing differential expression analysis at the single cell level using the Diffcyt algorithm [11]. Specifically, we analysed a publicly available mass cytometry dataset quantifying the immune cells in stimulated and unstimulated human peripheral blood cells (BCR_XL dataset [49], Additional file 1: Table S1). Notably, this is a paired experimental design, with each of the 8 independent samples, obtained from 8 different individuals, contributing to both stimulated and unstimulated

samples (16 samples in total). Importantly, this dataset was previously analysed using the Diffcyt algorithm to identify the cell state markers that were differentially expressed between the stimulated samples (BCR-XL group) and the unstimulated samples (Reference group). We refer readers to the Materials and Methods section for more information on the dataset. Our aim was to replicate these findings using a combination of SuperCellCyto and the Limma R package [40].

Our analysis workflow consists of steps 1, 2, 4, and 5 in Fig. 1B. First, we used SuperCellCyto (gamma = 20) to generate 8,641 supercells from 172,791 cells (step 1–2). We then annotated the supercells with the corresponding cell type labels (step 4) and calculated the mean expression of each cell state marker for every sample and cell type combination, akin to the pseudobulk approach commonly used in scRNAseq differential expression analysis. Next we used the Limma R package to test for expression differences between the stimulated and unstimulated samples, for each cell type separately, accounting for the paired experimental design (step 5, see Materials and Methods section for more details). Given the availability of cell type annotation, we slightly modified step 4 by annotating each supercell with the label of the most abundant cell type it encompasses, as opposed to clustering the supercells (refer to the Materials and Methods section for more details).

Our workflow successfully identified several cell state markers which show strong differential expression between the stimulated and unstimulated groups (False Discovery Rate (FDR) < = 0.05, Fig. 4A, B and Additional file 1: Fig. S11). Our findings were consistent with those identified by Diffcyt, including elevated expression of pS6, pPlcg2, pErk, and pAkt in B cells in the stimulated group, along with reduced expression of pNFkB in the stimulated group (Fig. 4A). We also recapitulated the Diffcyt results in CD4 T cells and Natural Killer (NK) cells, with significant differences in the expression of pBtk and pNFkB in CD4 T cells between the stimulated and unstimulated groups (Fig. 4B), and distinct differences in the expression of pBtk, pSlp76, and pNFkB in NK cells between the stimulated and unstimulated groups (Additional file 1: Fig. S11).

### Identification of differentially abundant rare monocyte subsets in melanoma patients

In this analysis, we investigated the capacity to conduct differential abundance analysis using supercells. We applied SuperCellCyto and Propeller [22] to a mass cytometry dataset quantifying the baseline samples (pre-treatment) of melanoma patients who subsequently either responded (R) or did not respond (NR) to an anti-PD1 immunotherapy (Anti_PD1 dataset [50], Additional file 1: Table S1). There are 20 samples in total (10 responders and 10 non-responders). The objective of this analysis was to identify a rare subset of monocytes, characterised as CD14 +, CD33 +, HLA-DRhi, ICAM-1 +, CD64 +, CD141 +, CD86 +, CD11c +, CD38 +, PD-L1 +, CD11b +, whose abundance correlates strongly with the patient's response status to anti-PD1 immunotherapy [11, 42].

Our analysis workflow consists of steps 1, 2, 3, 4, 6 in Fig. 1B. Firstly, we used SuperCellCyto (gamma = 20) to generate 4,286 supercells from 85,715 cells. We then used cyCombine to integrate the two batches together (Fig. 4C, D), and clustered the batch-corrected supercells using FlowSOM (see Materials and Methods for more details). We then identified the clusters representing the rare monocyte subset based on the

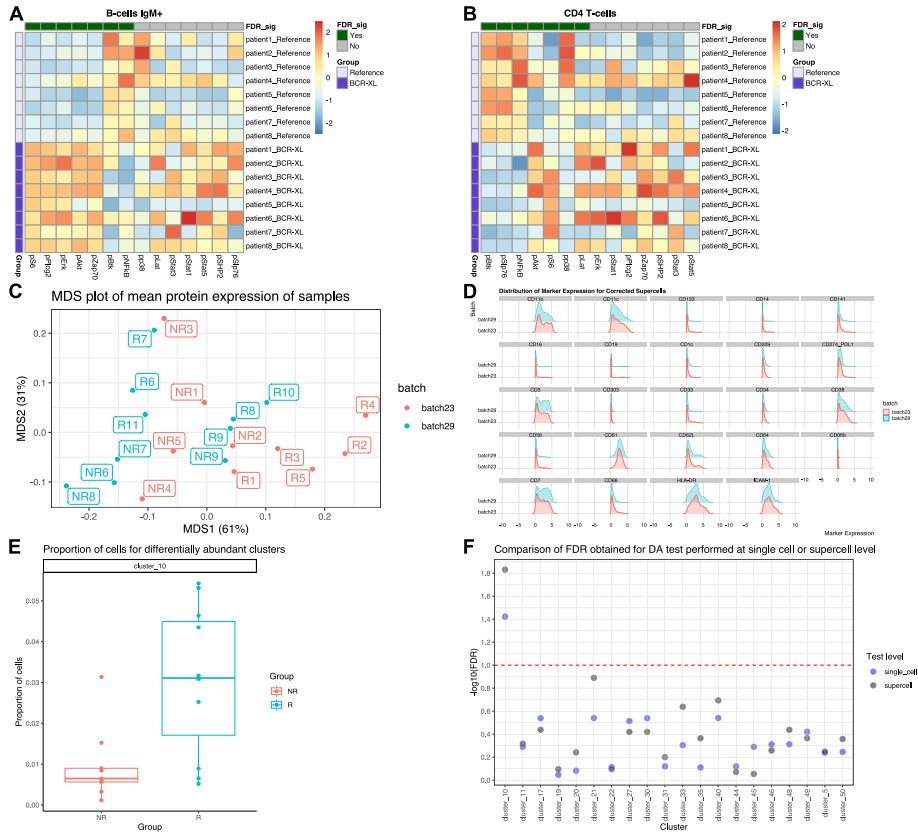

**Fig. 4** Differential expression and abundance analysis using supercells. **A**, **B** Heatmaps illustrate the scaled and centred median expression of cell state markers, calculated for each sample across the supercells, for B cells and CD4 T cells in the BCR_XL dataset. Each sample (row) is annotated according to its stimulation status, either with B cell receptor / FC receptor cross-linker (BCR-XL) or unstimulated. Each marker (column) is annotated based on its statistical significance as determined by Limma at a 5% False Discovery Rate (FDR_sig). **C-F** Differential abundance analysis following supercell creation for the Anti_PD1 dataset. **C** A Multidimensional Scaling (MDS) plot showcasing the variance in samples following cyCombine correction. **D** Distribution of marker expression for cyCombine-corrected supercells. **E** The proportion of single cells for the rare monocyte subset cluster (cluster_10). **F** Comparison of FDR obtained by running Propeller at the single cell or supercell level. The y-axis represents the -log10 transformed FDR, with lower FDR (more significant) corresponding to higher -log10 values. The red dotted line shows an equivalent FDR of 0.1

median expression of the aforementioned rare monocyte subset's signatory markers (see Additional file 1: Fig. S12A). For differential abundance analysis, we strongly recommend expanding the supercells back to single cells for more accurate cell type proportion calculation as each supercell contains different numbers of cells. Notably, expanding supercells back to single cells requires no additional computational effort. This is because SuperCellCyto readily provides a cell-to-supercell mapping which clearly identifies which supercell each cell belongs to. Once we expanded the supercells back to single cells, we retained only clusters that contained more than three cells from each sample, and performed a differential abundance test using Propeller, accounting for the batch. For comparison, we also applied Propeller directly on the supercells without expanding them back to single cells. For consistency, we only compared clusters that were retained by the filtering process.

Using the heatmap depicting the median expression of markers for each cluster (Additional file 1: Fig. S12A), we identified the rare monocyte subset as cluster 10. We found a statistically significant shift in abundance for cluster 10 between the responder (R) and non-responder (NR) groups, with responders exhibiting a higher proportion of these rare monocyte subsets (FDR < = 0.05, Fig. 4E).

Comparing the statistical test outcomes performed at the single cell and supercell level, we found that the difference in the abundance of supercells for the rare monocyte subset (cluster 10) was also statistically significant at the FDR threshold of 0.05 (Additional file 1: Fig. S12B, Fig. 4F). While this finding is consistent with the test outcome obtained from the single cell level abundance test, broadly, we observed variations in the FDR values obtained for each cluster (Fig. 4F). Neither the single cell nor the supercell level tests consistently yielded higher or lower FDR. Hence, our recommendation is to perform differential abundance analysis at the single cell level for the most accurate results.

### Efficient cell type label transfer between CITEseq and cytometry data

In this analysis, we investigated the potential for automating the annotation of cell types in cytometry data using previously annotated CITEseq data. Our workflow involves generating supercells from the cytometry data, subsetting the supercell and CITEseq data to only the common markers, and subsequently performing the label transfer process from the CITEseq data to the supercells, using either Seurat rPCA [16] or the Harmony alignment algorithm [51] combined with a kNN classifier (Harmony plus kNN). Supercells were only generated for the cytometry data, and not for the CITEseq data. For comparison, we also performed the cell type label transfer at the single cell level (single cell CITEseq to single cell cytometry). We demonstrate the effectiveness of the workflow by transferring the cell type annotation from the single cell CITEseq data generated by Triana et.al [52] to the supercells generated for the Levine_32dim mass cytometry data (see Additional file 1: Table S1).

We quantitatively assessed the effectiveness of label transfer using accuracy and weighted accuracy metrics. This involved aligning the cell type labels in the CITEseq data with those in the cytometry data. For each cell type in the cytometry data, we identified the corresponding cell type in the CITEseq data. In cases where the CITEseq data provided more granular subsets, we merged these subsets into broader categories. These broader categories were then matched with the equivalent cell type labels in the cytometry data. Specifically, for hematopoietic stem cells and progenitors, we combined the 3 subsets in the cytometry data into a group labelled CD34+_HSCs_and_HSPCs. We then mapped the subsets in the CITEseq data which expressed the CD34 transcript or antibody to this consolidated group. Cell types lacking direct counterparts, such as Basophils in the cytometry data or Conventional Dendritic Cells in the CITEseq data, remained unchanged. Further details about this mapping process is available in the Materials and Methods section. The resulting cell type label mapping, along with the UMAP plots showing both the original and mapped cell type annotation are available in the Additional file 1: Table S5 and Additional file 1: Fig. S13.

Using the cell type label mapping described above, for each true cell type label in the cytometry data, we calculated accuracy score. This score represents the

proportion of correctly labelled cells. To account for the different proportions of each cell type present in the cytometry data, we also computed a weighted accuracy score. We did this by multiplying the accuracy of each cell type by its relative proportion in the cytometry data (Additional file 1: Table S6). These weighted scores were then summed to produce an overall weighted accuracy score. Notably, cells not assigned true cell type labels were excluded from these calculations.

Figure 5A shows weighted accuracies for Seurat rPCA and Harmony plus kNN applied at supercell and single cell resolutions. Seurat rPCA outperformed Harmony plus kNN, achieving weighted accuracy of 0.71 (supercells) and 0.65 (single cells) against 0.67 (supercells) and 0.54 (single cells), respectively.

For both Seurat rPCA and Harmony plus kNN applied at the supercell level, we observed high accuracy scores (> 0.8) for Mature B cells, CD16 + NK cells, and CD8 T cells (Fig. 5B). A vast majority of the CD16 + NK cells were correctly identified as CD56dimCD16 + NK cells (Fig. 5C, Additional file 1: Fig. S14), while CD8 T cells were subdivided into various subsets, namely CD8 + CD103 + tissue resident memory T cells, CD8 + central memory T cells, CD8 + effector memory T cells, or CD8 + Naive T cells (Fig. 5C, Additional file 1: Fig. S14). Similarly, Mature B cells were broken down into Mature Naive B cells, Non-switched, or Class-switched Memory B cells. CD4 T cells were well annotated only by rPCA (accuracy of 0.91). Moderate accuracies (> 0.65) were obtained for Plasma B cells while low accuracies were noted for Pre B cells, pDCs, and CD16- NK cells. The latter is likely due to their limited representation in the CITEseq data (Pre B cells: 34 cells, 0.06%, CD16- NK cells: 597 cells 1.2%, pDCs: 568 cells 1.2%). No Basophils were correctly unannotated due to their absence in the CITEseq data.

Notable differences were observed between rPCA and Harmony plus kNN applied at the supercell level (Fig. 5B). For example, rPCA was better at identifying Pro B cells and CD4 T cells, while Harmony plus kNN was better at labelling CD34 + _HSCs_and_HSPCs.

When comparing the performance of Seurat rPCA and Harmony plus kNN at both supercell and single cell levels, we found both approaches demonstrated higher weighted accuracies when applied at the supercell level than at single cell level (Fig. 5A). However, this trend varied across different cell types (Fig. 5B). For example, CD16- NK cells exhibited higher accuracy at the single cell level as opposed to the supercell level. Interestingly, despite these differences, we noted that both rPCA and Harmony plus kNN yielded similar results in their classification patterns at both resolutions, as shown in Fig. 5C, Additional file 1: Fig. S14, Additional file 1: Fig. S15, Additional file 1: Fig. S16, and Additional file 1: Fig. S17.

In conclusion, while Seurat rPCA and Harmony plus kNN combined with SuperCellCyto are promising tools for assisting cell type annotation process, they should be considered as aiding the initial steps of the process, complementing rather than replacing manual annotation.

### Computational efficiency gains with SuperCellCyto

In this analysis, we examine the time taken to create supercells, and subsequently compare the time taken to perform several analysis steps, such as clustering, batch

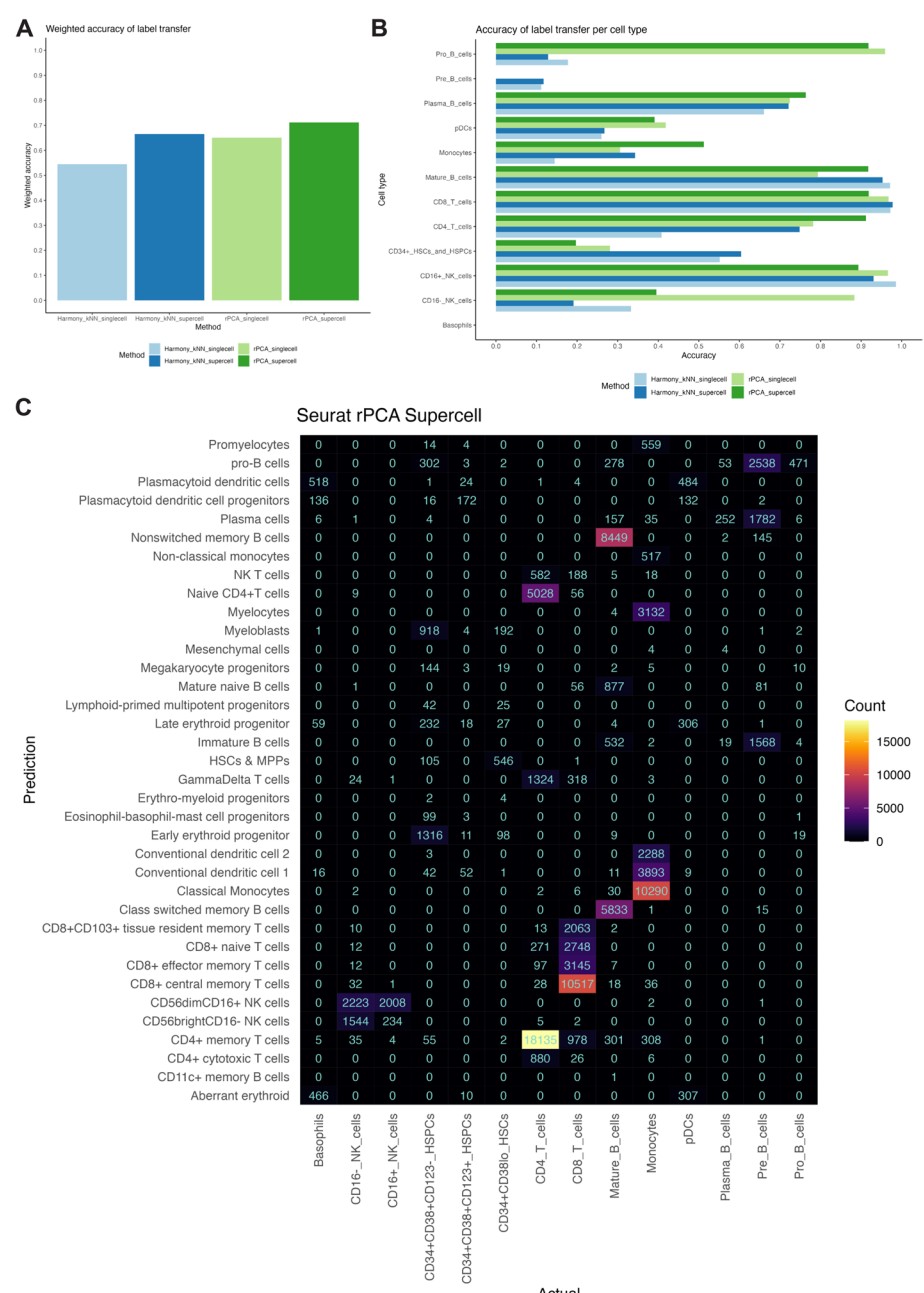

**Fig. 5** Performance evaluation of cell type label transfer from a CITEseq dataset to supercells. **A** Weighted accuracy of cell type label transfer. **B** Accuracy of label transfer for each cell type. **C** Confusion matrix of cell type label transfer performed by Seurat rPCA at the supercell level. Prediction (y-axis) represents the cell type label in the CITEseq data assigned using the label transfer method. Actual (x-axis) represents the cell type label in the cytometry data assigned using manual gating. Supercells were generated for the Levine_32dim cytometry dataset only. Label transfer was performed from single cell CITEseq data to the supercells generated for the cytometry dataset

correction, and cell type label transfer, on both supercells and single cells. For clustering and batch correction, we broadly explored the algorithms' parameter space. By contrasting these timings, we underscore the potential advantages of performing downstream analyses on supercells over single cells in terms of computational efficiency. We ran the methods on either a 2022 Macbook Pro (M2 chip, 24 GB RAM) or the High Performance Computing (HPC) platform using Nextflow [53] pipeline. The list of computing platforms used for each process is provided in the Additional file 1: Table S7.

Figure 6A illustrates the time taken to create supercells for 6 different cytometry datasets. These datasets contain a wide variety of number cells, ranging from tens of thousands to millions (Additional file 1: Table S1). Supercell creation for datasets containing less than one million cells took less than 3 min. For datasets comprising over 8 million cells, the time taken was 161 to 216 min.

For clustering Levine_32dim, Samusik_all, and Oetjen_bcells datasets (see Additional file 1: Table S1 for more details on the datasets) using Louvain, clustering supercells instead of single cells resulted in significant time savings (Fig. 6B). For Levine_32dim, clustering supercells took < 1 min while clustering single cells took 4 to 19 min. For the Samusik_all datasets, clustering supercells took < 5 min while clustering single cells took

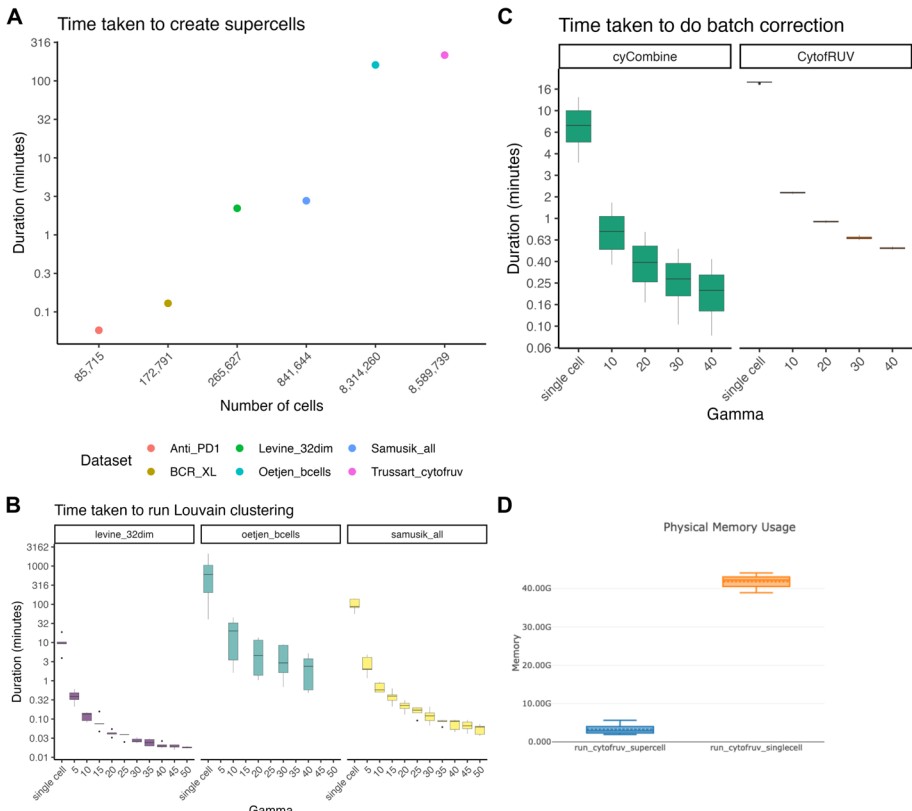

**Fig. 6** Evaluating the run time of SuperCellCyto and subsequent downstream analyses. **A** The runtime of SuperCellCyto for six cytometry datasets, measured in minutes. **B** The runtime of Louvain clustering for Levine_32dim, Oetjen_bcells, and Samusik_all datasets, measured in minutes. **C** The runtime of cyCombine and CytofRUV for correcting batch effects in Trussart_cytofRUV dataset. For both (**B**) and (**C**), x-axis denotes whether the clustering was done on single cells or supercells created using various gamma values. **D** The memory consumption of CytofRUV for correcting batch effects in Trussart_cytofRUV dataset either at the single cell (orange) or supercell level (blue)

56 to 140 min. For the Oetjen_bcells dataset, which comprises over 8 million cells, we observed stark differences. Clustering single cells took 41 min to 35 h, whereas clustering supercells took 0.5 to 45 min.

Similarly, for FlowSOM clustering, we also observed improvement in run time for both the Levine_32dim and Samusik_all datasets when clustering supercells (Additional file 1: Fig. S18).

Comparing the time taken to run batch correction (Fig. 6C) on the Trussart_cytofruv dataset (Additional file 1: Table S1), performing batch correction at the supercell level took less than 2 min, whereas correcting batch effects at the single cell level could take up to 19 min. Importantly, we observed a 14 fold reduction in the amount of memory taken to run CytofRUV at the supercell level compared to the single cell level (~3 GB vs ~42 GB, Fig. 6D).

Lastly, transferring cell type labels from CITEseq data to supercells generated for a cytometry dataset resulted in 4 to 5 fold improvement in run times. Specifically, transferring these labels to supercells generated for the Levine_32dim cytometry data took 9.6 min with Seurat rPCA and 1 min with Harmony plus kNN. On the other hand, transferring these labels to the single cells in the same dataset (Levine_32dim) took 37 min using Seurat rPCA and 5.4 min with Harmony plus kNN.

## Discussion

The increasing size and complexity of cytometry data has rendered traditional manual gating inadequate, necessitating the adoption of computational methods for data analysis. A myriad of computational methods designed for cytometry and scRNAseq data analysis offer robust, data-driven, and diverse analysis workflows. However, many of these tools, particularly those developed for scRNAseq, often require extensive processing time due to the vast number of cells typically found in cytometry data. When algorithm parameters need to be adjusted and analysis workflows repeated, the computational time required to analyse large cytometry datasets with millions of cells can increase to days and even weeks.

To address this challenge, we propose a strategy for reducing large cytometry datasets that can drastically reduce computational time from hours to seconds. We developed an R package, SuperCellCyto, which leverages and extends the SuperCell R package to aggregate phenotypically similar cells based on their marker expressions. We further enhanced this approach by introducing the ability to process multiple samples concurrently using a custom load balancing strategy. Utilising this framework, we generated supercells for six publicly available cytometry datasets and successfully performed various downstream analyses. These included clustering, cell type annotation, differential expression and abundance analysis, and batch correction. Moreover, we demonstrated the feasibility in using SuperCellCyto in conjunction with Seurat rPCA and Harmony combined with a kNN classifier to streamline the cell type annotation process. Our findings underscore the feasibility and effectiveness of our proposed strategy in managing the increasing complexity of cytometry data analysis.

In cytometry, markers are typically classified into two categories: cell type markers and cell state markers. Cell type markers serve to identify distinct cell types, whereas cell state markers are employed to investigate cell states. For most analyses, we recommend

generating supercells using all available markers, as opposed to exclusively using cell type markers. This approach minimises the risk of creating supercells that encompass cells of identical type but differing states, thereby enhancing the reliability and accuracy of downstream analyses. Importantly, SuperCellCyto offers the flexibility to generate supercells based on either cell type markers or cell state markers exclusively, catering to the specific requirements of different analyses.

The granularity of supercells is governed by the gamma parameter. A larger gamma value results in the generation of fewer supercells, each potentially encompassing a greater number of cells, whereas a smaller gamma value yields more supercells, each containing fewer cells. The selection of an appropriate gamma value is contingent upon the specific analysis objectives and the desired level of data compression. Generally, we found a gamma value of 20 strikes a good balance between preserving cell type diversity and reducing data size.

One key advantage of using SuperCellCyto for an analysis is that the creation of supercells will typically only need to be run once at the outset. This allows subsequent downstream algorithms, which often need to be run multiple times with varying parameters, to benefit from the performance gains offered by supercells. If, after creating supercells once, there is a need to change the number of supercells produced, there is no need to rebuild the kNN graph or rerun the Walktrap algorithm. We provide a function within SuperCellCyto which will recut the dendrogram produced by Walktrap algorithm based on a new gamma parameter.

Importantly, for all analyses in this study, we ran SuperCellCyto on a 2022 Macbook Pro laptop equipped with the M2 chip and 24 GB of RAM. Future research could explore strategies to further improve the processing time required to generate supercells. This could involve accelerating certain internal calculations by leveraging the multicore processing capabilities commonly available in contemporary computers or harnessing the power of Graphics Processing Units (GPUs).

We have successfully demonstrated the ability to mitigate batch effects at the supercell level using CytofRUV and cyCombine. However, it is important to note that batch correction executed at the supercell level does not translate to correction at the single cell level. Consequently, for downstream analyses that compare marker expression and necessitate the expansion of specific supercells back to the single cell level, batch correction must be reapplied to these single cells.

When performing differential abundance analysis, we noted slight differences in results when running Propeller at the supercell and single cell levels. These differences could potentially be attributed to the fact that each supercell does not encapsulate an identical number of cells. The gamma parameter primarily dictates the number of supercells generated rather than the number of cells encompassed within each supercell. Consequently, we advocate for the expansion of supercells back to single cells prior to undertaking differential abundance analysis for the most accurate and reliable analysis.

We observed that in certain cases, the time savings from processing supercells as opposed to single cells can be marginal, for instance when clustering supercells using FlowSOM. However, it is crucial to recognise that not all algorithms (or parameter settings) exhibit the same level of efficiency. Many can substantially benefit from processing supercells instead of single cells. For instance, when running Louvain clustering, the

time taken to cluster supercells was significantly less than the time taken to cluster single cells. Similarly, we also observed a significant reduction in memory consumption when correcting batch effects at the supercell level compared to single cells using cytofRUV.

In SuperCellCyto, supercells are generated using single cell kNN graphs combined with the Walktrap algorithm. However, various other types of graphs and community detection algorithms are available. Future work could investigate a plug and play functionality which allows users to select their preferred type of graph to use (e.g., shared k nearest neighbour graphs or self organising map) and subsequently the algorithms to identify densely connected subgraphs (e.g., Louvain, Leiden, or hierarchical clustering). Notably, the SuperCell R package is not the only approach currently available for creating supercells, or metacells, from scRNAseq data. Future research could explore and compare other methods, such as SEAcells [30], MetaCell [28, 29], and GeoSketch [54], with a specific focus on their efficacy in compressing cytometry data.

Lastly, in our assessment, we demonstrated the ability to use Seurat rPCA and Harmony in conjunction with a kNN classifier to aid the cell type annotation process. Importantly, the inherent complexities and nuances in cell type identification still necessitate a degree of manual expertise and judgement. Therefore, while these methods provide valuable insights and can significantly streamline the annotation process, they function optimally as precursors to, rather than substitutes for, manual annotation by experts in the field. Notably, this area presents substantial opportunities for further exploration and development. Recent promising new methods, such as StabMap [55], Seurat CCA [17], and MultiMAP [56], could potentially further improve cell type label transfer between cytometry and CITEseq data.

## Conclusions

Advancements in cytometry technologies have led to rapid increase in the size and complexity of cytometry data. These datasets now routinely encompass up to 50 proteins and millions of cells, presenting monumental scalability challenges as many existing tools struggle with extensive run times when processing such large high-dimensional datasets. SuperCellCyto addresses this by grouping cells with similar marker expression into supercells. It effectively reduces dataset size while preserving biological heterogeneity, thus significantly alleviating the computational demands of downstream analysis. SuperCellCyto is available as an open source R package on GitHub (https://github.com/phips onlab/SuperCellCyto) [32] and Zenodo (https://zenodo.org/records/10521294) [33] and can be seamlessly integrated into existing analysis pipelines.

## Materials and Methods

### Datasets

The Levine_32dim dataset is a mass cytometry data that profiles healthy human bone marrow. It comprises two samples and 32 markers, with a total of 265,627 cells. Of these cells, 39% (104,184) were assigned 14 cell types using manual gating.

The Samusik_all dataset is a mass cytometry data that profiles healthy mice bone marrow. It includes 10 samples and 39 markers, with a total of 841,644 cells. Of these, 61% (514,386) were assigned 24 cell types using manual gating.

The Oetjen_bcell data profiles healthy human bone marrow using flow cytometry. The dataset contains 13 markers, 22 samples, and 18,648,656 events. While no cell type annotation is provided for each cell, the manual gating scheme previously used by the authors to identify the B cell subsets was made publicly available and subsequently used to label the clusters. The manual gating scheme is available in the Additional file 1: Note S1.

The BCR_XL dataset consists of 8 paired samples of healthy human peripheral blood cells. These samples are either stimulated with B cell receptor / FC receptor cross-linker (BCR-XL) or left unstimulated [49].

The Anti_PD1 dataset quantifies immune cell subsets in the peripheral blood of melanoma patients subsequently treated with anti-PD1 immunotherapy. It contains 20 samples taken from melanoma patients prior to undergoing anti-PD1 immunotherapy treatment. The samples were divided into two groups: those who subsequently responded to the treatment (responder or R) and those who did not (non-responder or NR), and quantified across two batches (batch 23 and batch 29). There are, in total, 11 samples from the responder group, and 9 samples from the non-responder group.

The Trussart_cytofruv dataset quantifies the peripheral blood mononuclear cells of 9 healthy individuals and 3 Chronic Lymphocytic Leukaemia patients (CLL). The dataset contains 12 paired samples (1 paired sample per patient) processed over 2 batches, with each batch containing 1 sample from every patient. The dataset includes 31 markers, specifically 19 cell type markers and 12 cell state markers.

The CITEseq dataset profiles the mononuclear bone marrow cells from hip aspirates of healthy young and old adults using 97 antibodies and targeted sequencing. The dataset includes 10 samples and 49,057 cells.

## Analysis workflows
### Data transformation
All mass cytometry datasets were transformed using an inverse hyperbolic sine (arc-sinh) transformation with a co-factor of 5 before the generation of supercells or any subsequent downstream analyses. Similarly, the flow cytometry dataset underwent arc-sinh transformation with a co-factor of 150. For the CITEseq data, a Centred Log Ratio transformation was used to transform the protein expression.

### Cell type identification for annotated mass cytometry data
For the Levine_32dim and Samusik_all datasets, SuperCellCyto was used to generate supercells, using gamma values that ranged from 5 to 50, in increments of 5. Following this, the supercells were clustered using the FlowSOM and Louvain clustering algorithms. An extensive exploration of the parameter space was conducted for both clustering algorithms. Specifically, for FlowSOM, the grid size and the number of meta-clusters were varied, while for Louvain, the parameter k, which governs the number of nearest neighbours considered when constructing the single cell network graph, was varied. The specific range of values explored for each parameter is documented in Additional file 1: Table S3.

### Cell type identification for flow cytometry data

For the Oetjen_bcell dataset, an initial step of manual gating was performed using Cyto-Explorer [57] to exclude debris and doublets. This resulted in over 8.3 million single live cells (refer to Additional file 1: Fig. S19 for the manual gating scheme). Following this, SuperCellCyto was used to generate supercells, with the gamma parameter set to 20, yielding 415,711 supercells. These supercells were subsequently clustered using the Louvain clustering algorithm with the parameter k set to 3, based on the expression of markers used in the manual gating strategy by Oetjen et.al to identify the B cell subsets (see Additional file 1: Note S1) [38], namely CD19, CD45, CD10, CD20, CD27, CD21, CD38, and CD138. Clusters were then annotated with B cell subsets based on the manual gating scheme provided in the original publication of the datasets [45] (see Additional file 1: Note S1). Approximately 32% of the supercells expressed CD45 and CD19. The remaining supercells were CD45- and CD19- (68%) and were excluded from Fig. 2D, E and Additional file 1: Fig. S6B.

### Differential expression and abundance analysis

For the BCR_XL dataset, SuperCellCyto (with the gamma parameter set to 20) was used to generate 8,641 supercells. Using the cell type annotations provided with the dataset, the supercells were annotated based on the cell type that was most abundantly represented within each supercell. Subsequently, the mean expression of all cell state markers was computed for each combination of sample and cell type. Following this, Limma was applied to each cell type. Specifically, a linear model was first fitted using Limma's *lmFit* function, followed by the application of Limma's *eBayes* and *treat* (*fc* parameter set to 1.1) functions.

For the Anti_PD1 dataset, SuperCellCyto was used (with the gamma parameter set to 20) to generate 4,286 supercells. These supercells were then batch corrected using cyCombine (grid size set to $4 \times 4$) clustered using FlowSOM using $20 \times 20$ grid size and 50 metaclusters. We then performed a differential abundance test (taking the batch into account) using Propeller. Propeller was run either on the single cells expanded from the supercells or directly on the supercells. Importantly, we tested only clusters which captured more than 3 cells from each sample for both the single cell and supercell level tests.

### Batch correction

For the Trussart_cytofruv dataset, an initial preprocessing step was carried out using the CATALYST R package and the R scripts provided with the dataset [46]. Following this, SuperCellCyto was employed (with the gamma parameter set to 20) to generate 429,488 supercells from over 8.5 million cells.

Subsequently, CytofRUV (with parameter k set to 5 and samples from CLL2 and HC1 patients used as the pseudo-replicates) [46] and cyCombine (with grid size set to $8 \times 8$) [47] were run, to integrate the two batches. CytofRUV is a Remove Unwanted Variation (RUV)-based method [58], specifically the RUV-III method [59], for identifying and removing unwanted technical variations such as batch effect. CyCombine [47] is a cytometry data integration method based on the Empirical Bayes method ComBat [60].

### Cell type label transfer

SuperCellCyto, with a gamma parameter set to 20, was used to generate 13,282 supercells for the Levine_32dim dataset. Both the CITEseq data and the supercells were subsequently subsetted to retain only the markers common to both, resulting in a total of 21 shared markers.

The datasets were then processed by Seurat rPCA which first identified the transfer anchor between the two datasets, and thereafter transferred the cell type annotation from the CITEseq data to the supercells.

In the case of the Harmony and kNN approach, Harmony was employed to align the CITEseq data and the supercells. Following this, a k-Nearest-Neighbour classifier (with the k parameter set to 1) was trained on the CITEseq data and subsequently used to classify the supercells.

Finally, each supercell was expanded, and the cell type label of the corresponding supercell was assigned to the resulting single cells. Importantly, for evaluation purposes, only the cells for which manually gated cell type annotations were available were included in the analysis.

*Mapping the cell type label between cytometry and CITEseq data*   To calculate accuracy and weighted accuracy scores, we needed to reconcile cell type labels between the cytometry and CITEseq data. Direct one-to-one label correspondences existed for CD16-_NK_cells, CD16+_NK_cells, pDCs, Plasma_B_cells, Pre_B_cells, and Pro_B_cells.

In instances where the CITEseq data provided more granular cell type labels than the cytometry data, specifically for CD4_T_cells, CD8_T_cells, and Monocytes, we amalgamated the various subsets into broader categories. For example, in the CITEseq data, subsets such as CD4+cytotoxic T cells, CD4+memory T cells, and Naive CD4+T cells were combined into one group and mapped to the CD4_T_cells label that correspond to CD4+T cells in the cytometry data.

Similarly, for the Mature_B_cells label in cytometry data, we consolidated Mature B cells subsets in CITEseq data, including CD11c+memory B cells, Mature naive B cells, Class switched memory B cells, and Nonswitched memory B cells, into one group, and mapped it to the Mature_B_cells label. These 4 subsets were selected based on their CD19, IgD, and IgG expression (Additional file 1: Fig. S13D).

For hematopoietic stem cells and progenitors, we grouped the 3 subsets present in the cytometry cytometry data, namely CD34+CD38+CD123-_HSPCs, CD34+CD38+CD123+_HSPCs, and CD34+CD38lo_HSCs into a single group named 'CD34+_HSCs_and_HSPCs'. Subsequently, all stem cell and progenitor subsets in the CITEseq dataset expressing CD34 (Additional file 1: Fig. S13C) were mapped to this unified label.

Cell types without clear mappings, such as Basophils in cytometry data or Conventional Dendritic Cells in CITEseq data, were left unchanged.

The resulting mapping used for calculating accuracy and weighted accuracy is provided in Additional file 1: Table S5. UMAP plots of the CITEseq data, depicting original and mapped cell type labels, along with the expressions of CD19, IgD, and CD34, are available in Additional file 1: Fig. S13.

*Runtime benchmarking*

The computing platforms used for each process is provided in the Additional file 1: Table S5. Nextflow pipelines were used to run the processes on the High Performance Computing platform. The amount of RAM and CPUs allocated are specified within the Nextflow scripts available on https://github.com/phipsonlab/SuperCellCyto-analysis [35].

Runtimes were measured using the *tictoc* R package [61]. Each operation was repeated twice to ensure consistency, and the mean duration was reported.

## Evaluation metrics

### *Purity and adjusted rand index*

To evaluate the quality of the generated supercells and the results of the clustering process, two metrics were used: purity and the Adjusted Rand Index (ARI). The purity metric quantifies the degree to which each supercell is composed of a singular cell type, with values spanning from 0 to 1. A purity value of 1 indicates that a supercell is entirely composed of cells from a single cell type.

ARI measures the agreement between the clustering results and a known ground truth, taking into account the influence of chance. ARI value can range from -1 to 1, with a score of 1 signifying maximum agreement between the clustering results and the ground truth. Conversely, ARI values of 0 and -1 suggest that the agreement is either the same or worse than what would be achieved by random chance.

Notably, purity and ARI scores were computed only for cells for which cell type annotations were available, which constituted 39% of cells for the Levine_32dim dataset and 61% of cells for the Samusik_all dataset.

To compute the ARI score, each supercell was expanded to the single cell level, ensuring that each cell within a given supercell was assigned the cluster label of the supercell. Finally, for both datasets, two types of ground truths were used: the cell type annotation provided with the dataset, and the clustering results obtained by clustering the single cells using the same combination of parameter values.

### *Earth mover distance*

The Earth Mover's Distance (EMD) metric was employed to evaluate the efficacy of batch effect correction algorithms by quantifying the dissimilarity in the distribution of markers across different batches. Following a successful batch correction, the EMD score for any given marker in the dataset is anticipated to be lower than the score obtained prior to the application of the batch effect correction algorithm.

In our evaluation, the EMD score was calculated for each paired sample at the supercell level. Initially, the distribution of each marker was obtained by binning the data into bins of size 0.1. Subsequently, for each marker and paired samples, the EMD score was computed to compare the differences in the distribution of the marker values.

### Scib metrics

NMI (Normalised Mutual Information) compared the overlap of clusterings obtained before and after batch correction. Clustering was done using the FlowSOM [6] algorithm with the number of metaclusters set to 20 and the grid size set to $10 \times 10$. We used scib's implementation of NMI which scales the metric's value between 0 (indicating no overlap between clustering labels) and 1 (denoting perfect overlap), based on the entropy of the cluster labels.

ARI measured the similarity between the clustering results obtained before and after batch correction. Clustering was performed using FlowSOM on both uncorrected and corrected supercells, maintaining the same settings of 20 metaclusters and a $10 \times 10$ grid. ARI values range from 0, signifying random labelling, to 1, indicating a perfect match between the clustering results.

ASW_label (Average Silhouette Width) evaluated the compactness and separation of clusters using silhouette width metric. We used scib's implementation [48] of ASW_label which scales its values to between 0 (worst) and 1 (best). ASW_label was calculated on the PCA embeddings of the supercells with the number of principal components (PCs) set to 20. Due to the absence of cell type annotations in the Trussart_cytofruv dataset, we generated cluster labels using FlowSOM, clustering both batch-corrected and uncorrected supercells with metaclusters set to 20 and grid size set to $10 \times 10$. Subsequently, we used these labels as proxies for cell type labels.

## Supplementary Information

---

**Additional file 1.** SuperCellCyto: enabling efficient analysis of large scale cytometry datasets. Supplementary file containing Note S1, Table S1-S7, and Fig. S1-S19.

**Additional file 2.** Review history.

---

#### Acknowledgements
We thank Prof. Gordon Smyth, Dr. Nadia Davidson, Dr. Feng Yan, Ms. Mengbo Li, and all members of the Phipson Lab at the Walter and Eliza Hall Institute of Medical Research (WEHI) for feedback on the downstream analysis, the WEHI Research Computing Platform for providing access to their High-Performance Computing facility, A/Prof. Matt Field at the James Cook University, A/Prof. Anne Bruestle and Mr. Tony Xu at the Australian National University for feedback and discussion on supercells, and the attendees of joint Australian Society of Immunology/Oz Single Cell 2022 Hackathon for catalysing this work.
This publication is part of the Human Cell Atlas – humancellatlas.org.

#### Peer review information

#### Review history
The review history is available as Additional file 2.

#### Authors' contributions
GHP and BP designed the study. GHP, GH, and BP developed the package with input from FMW and TMA. GHP performed the data analysis with input from FMW, TMA, and BP. GHP and BP wrote the manuscript with input from all authors. BP supervised the study. All authors read and approved the final manuscript.

#### Funding
This work was supported by a National Health and Medical Research Council Investigator grant [GNT1175653] to B.P., and the International Society for the Advancement of Cytometry (ISAC), Marylou Ingram Scholars Program to F.M-W. The joint Australian Society of Immunology/Oz Single Cell 2022 Hackathon which initiated the study was supported by the Australian Society for Immunology, Oz Single Cell, and the Chan Zuckerberg Initiative.

#### Availability of data and materials
The SuperCellCyto R package is publicly available on GitHub (https://github.com/phipsonlab/SuperCellCyto) [32] and Zenodo (https://zenodo.org/records/10521294) [33], along with extensive vignettes (https://phipsonlab.github.io/SuperCellCyto/) [34], providing a valuable resource for the research community to incorporate SuperCellCyto into their analysis

pipelines more efficiently and effectively. Additionally, the complete analysis workflow associated with the results presented below can be accessed online on https://github.com/phipsonlab/SuperCellCyto-analysis [35].

The SuperCellCyto implementation is available as an R package on GitHub (https://github.com/phipsonlab/SuperCellCyto) [32] and Zenodo (https://zenodo.org/records/10521294) [33] under GNU General Public License v3.0. Vignettes, function documentations, contributing guidelines, and installation instructions are available on https://phipsonlab.github.io/SuperCellCyto/ [34]. SuperCellCyto v0.1.0 (https://doi.org/https://doi.org/10.5281/zenodo.10521294) [33] was used in this manuscript.

All analysis performed in this paper is available as a workflowr [62] website at https://phipsonlab.github.io/SuperCellCyto-analysis/ [35], with the original source code available on GitHub (https://github.com/phipsonlab/SuperCellCyto-analysis) [35].

All datasets used in the study have previously been published, with references available both within the article and in Additional file 1: Table S1. Data files required to reproduce the analyses are available on Zenodo (https://doi.org/https://doi.org/10.5281/zenodo.8274907) [63]. Anti_PD1, BCR_XL, Levine_32dim, and Samusik_all data were downloaded using the HDCytoData R package [64] version 1.18.0. CITEseq data was downloaded from CellxGene [65].

## Declarations

### Ethics approval and consent to participate
Not applicable.

### Consent for publication
Not applicable.

### Competing interests
The authors declare that they have no competing interests.

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

## 