## [**Additional file 2.** Review history. · Genome Biology]

Review History

First round of review

Reviewer 1

Were you able to assess all statistics in the manuscript, including the appropriateness of statistical tests used? There are no statistics in the manuscript.

Were you able to directly test the methods? Yes.

Comments to author:

The authors present SuperCellCyto, a software extending on the SuperCell R package, and demonstrated its applicability to several types of cytometry data (flow, cyTOF and CITEseq). The main focus of the manuscript is to demonstrate that reducing the complexity of the data by aggregating single-cells into SuperCells preserves rare cell types and that the approach is largely compatible with the analysis workflow for cytometry data, such as clustering, batch effect correction, differential marker intensity analysis, differential abundance analysis, and automated cell type annotation. Overall, the authors reasonably argue that using SuperCells is a valid approach to reduce complexity in cytometry data. However, the showcases for batch effect correction and label transfer do not adequately support the conclusions drawn by the authors.

The software itself can be installed from GitHub and has extensive dependencies. I could not install it on my MacBook Air (2022) due to issues with the RSpecra dependency in other packages, but I successfully installed it on a Google Colab instance, which took about 30 min to complete for SuperCell and another min for the SuperCellCyto package. The package seems to be well written and extensively documented, with all the code used for reproducing the figures available online.

The software builds mostly on the previously published SuperCell R package. The extension in SuperCellCyto allows to shift the granularity of the SuperCells without recomputing the entire SuperCell object. The novelty of the approach seems limited to me: The authors mostly explore the realm of cytometry data instead of single-cell RNA-sequencing data with their tool. The idea of coarse-graining cytometry data while avoiding that rare cell types succumb to the subsampling process has been explored previously with HSNE (see van Unen, V., Höllt, T., Pezzotti, N. et al. Visual analysis of mass cytometry data by hierarchical stochastic neighbour embedding reveals rare cell types. Nat Commun 8, 1740 (2017) and the algorithmic basis in Pezzotti, N. et al. Approximated and user steerable tSNE for progressive visual analytics. IEEE. Trans. Vis. Comput. Graph. 23, 1739-1752 (2016)). The HSNE approach has been adopted by a broad audience of biologists, mostly because it preserves the original number of cells and topology, as opposed to random subsampling. I see the potential that the concept of SuperCells gains a similar popularity.

The authors demonstrate the plausibility of the SuperCell aggregation approach to cytometry data, but do not compare to alternative approaches like HSNE, or a simple baseline of random subsampling. The aggregation procedure is insufficiently described: While it is reasonable to sum count data in single-cell RNAseq, it is not clear how the authors address the aggregation process for marker intensities, which usually include negative values. Also, summing after arcsinh/logicle-scaling has a different mathematical meaning than summing count data. Also, is

aggregation handled differently for CITE-seq data, which is count based? I think that the authors should address this issue in more detail.

When addressing the purity of the SuperCells, the authors use the metric called purity (which is related to the term precision in machine learning). Instead of purity, one could also use the local inverse Simpson index (LISI) to determine the effective number of cell types in each of the supercells (see Korsunsky, I., Millard, N., Fan, J. et al. Fast, sensitive and accurate integration of single-cell data with Harmony. *Nat Methods* 16, 1289-1296 (2019)). The metric purity and the cutoff chosen at 0.5 does not fully support the statement that most supercells consist of one or predominantly one cell type. What about the supercells with a purity between 0.5 and 0.9?

Batch effect correction presented in Figure 3: The authors show an example for batch correction, where they regress out the batch effect for 12 samples with approx. 400,000 supercells. While they aim to demonstrate the effectivity of the batch effect correction, they do not sufficiently demonstrate the preservation of the biological signal in their test. The visualization of the sample distribution by batch using MDS indicates a trend, but does not show whether the same cell types are accurately integrated. There is an extensive body of literature for the integration of single-cell RNA-seq and ATAC-seq data, including a variety of metrics to quantify both batch effect removal and preservation of the biological signal (see for instance Luecken, M.D., Büttner, M., Chaichoompu, K. et al. Benchmarking atlas-level data integration in single-cell genomics. *Nat Methods* 19, 41-50 (2022) for an extensive set of suitable and scalable metrics).

What is the purity of the supercells in terms of batch effect? Does every supercell contain only cells of the same batch/sample or is there a mixing? If yes, what is the degree of mixing?

Using the EMD for the marker distribution is great, but also does not control for over-correction. When all data is squished into one point instead of a distribution, EMD would be 0. The authors should provide a reasonable baseline that accounts for overcorrection.

The label transfer of CITEseq to cyTOF needs extensive revision in both data representation and evaluation of the resulting labels. Figures 5, Supplementary Fig. S9/S10 shouldn't be a dot plot where both the color scheme and the dot size represent the same quantity. A more meaningful representation is a confusion matrix. Second, the prediction using Harmony and a kNN classifier looks pretty inaccurate (no distinction in NK cell subsets even though the class label is present, pro_B_cells annotated as erythroid progenitor cells). The picture is a little better for Seurat rPCA (single-cell, S10), but it failed to predict monocyte class accurately. I personally do not believe that these two methods have a similar accuracy. I suggest that the authors compute the weighted accuracy per cell type to reduce the bias of accurate labeling of the most frequent cell type. Furthermore, the authors should visualize the transferred labels e.g. by showing a UMAP of the annotated data in addition. Overall, I do not agree with the author's conclusion that the automated cell type classification from CITEseq to cytometry data worked sufficiently well to put this into production. It is rather a starting point for manual cell type annotation.

Differential abundance testing: I appreciate pointing out that using the supercells to compute cell type proportions distorts the overall proportions. What is the computational effort to expand the supercells?

Minor Comments:

Figure 2A: one could integrate the mean purity plot with Figure 2B and visualize the mean as a single dot in each distribution plot.

Regarding figures 2A and B: What is the size distribution of the supercells as a function of gamma (i.e. how many cells aggregate into a supercell)? What is the expected purity level compared to random sampling?

Figure 2F: Indicate in the figure caption the dataset used for this plot (I assume it's the Oetjen B cell dataset.) Please indicate the individual proportions per sample - a box plot might represent the data better.

Time and memory efficiency (Figure 6): Please indicate the type of compute infrastructure where you conducted your tests on. It might be very different for a linux server node with 128 GB memory and 32 cores than for a laptop computer.

Reviewer 2

Were you able to assess all statistics in the manuscript, including the appropriateness of statistical tests used? Yes, and I have assessed the statistics in my report.

Were you able to directly test the methods? Yes.

Comments to author:

The manuscript "SuperCellCyto: enabling efficient analysis of large scale cytometry datasets" navigates the rapidly evolving landscape of cytometry technologies, which now enables the quantification of an impressive 50 proteins across millions of cells at single-cell resolution. Acknowledging the indispensable role of computational tools in the analysis of such expansive datasets, the authors articulate a critical issue faced by existing methods—their susceptibility to prolonged run times when confronted with the vast scale of cytometry data. Moreover, the challenge of handling large datasets is compounded by the inadequacies of conventional solutions, such as random subsampling, which may inadvertently exclude small, yet crucial, rare cell subsets.

To address these formidable challenges, the authors propose an innovative strategy that draws inspiration from the SuperCell framework in the single-cell RNA sequencing (scRNAseq) domain. The supercell concept, involving the grouping of single cells with highly similar transcriptomic profiles, has proven its efficacy in scRNAseq data analysis. Remarkably, the manuscript provides compelling evidence that this concept seamlessly extends to cytometry datasets, eliminating concerns about information loss when grouping cells into supercells.

The manuscript not only introduces this novel strategy but also substantiates its effectiveness through a series of downstream analyses on six publicly available cytometry datasets at the supercell level. The authors successfully replicate previous findings conducted at the single-cell

level, showcasing the robustness and applicability of their proposed approach. A noteworthy contribution lies in the presentation of a computationally efficient solution within the SuperCellCyto R package, designed for the seamless transfer of cell type labels from single-cell multiomics data, combining RNA with protein measurements, to cytometry datasets. This advancement holds promise for elevating the precision of cell type annotations in cytometry studies.

In summary, the manuscript encapsulates a pioneering effort to address critical bottlenecks in large-scale cytometry data analysis. The proposed strategy, supported by compelling evidence and facilitated through an accessible R package, positions this work as a valuable contribution to advancing the methodologies that underpin our understanding of cellular heterogeneity and function.

Minor Feedback

- 1) The tool could become part of the analysis routine for many biologists with bioinformatics skills. However, the paper does not specify the minimum requirements for execution and the number of cells tested (RAM, Number of processors).
- 2) I suggest creating a page on <https://about.readthedocs.com/?ref=readthedocs.com>, incorporating all information related to the new package, such as "Installation," "Tutorials," "Usage Principles," "Release Notes," and "Contributing." Additionally, improving the documentation of the tool is recommended.
- 3) The tutorial content accompanying the tool under consideration is commendable in its clarity and effectiveness, particularly in guiding users through the testing of various functions with datasets. However, to further enhance the utility of the tool, I recommend the inclusion of comprehensive tutorials addressing data preparation across different scenarios. Specifically, it would be immensely beneficial for users if the tutorial provided guidance on data preparation starting from various data formats such as FCS files or CSV files, and also encompassed more specialized formats commonly used in single-cell analysis, such as SingleCellExperiment or Seurat objects.

Starting from FCS or CSV Files:

Begin with a step-by-step guide on how to preprocess and prepare data when starting from raw FCS (Flow Cytometry Standard) files. Address common preprocessing steps, quality control measures, and any necessary transformations for compatibility with the tool.

Extend the tutorial to cover a similar workflow for users who prefer working with CSV files, providing insights into the required formatting and pre-processing steps.

Utilizing Classical Single-Cell Data Formats:

For users working with established single-cell data formats like SingleCellExperiment or Seurat objects, create a tutorial segment explaining how to seamlessly integrate these formats with the tool.

Detail any specific considerations or transformations necessary to ensure smooth compatibility, while also highlighting the advantages of leveraging these widely-used formats in conjunction with the tool.

By incorporating tutorials that address these diverse data preparation scenarios, the tool's user base can benefit from a more inclusive and versatile resource. Not only will this approach cater to users at different stages of data analysis proficiency, but it will also serve as a valuable reference for researchers working with various types of single-cell data. This addition will undoubtedly enhance the tool's accessibility and usability, aligning it more closely with the diverse needs of the user community.

Response to reviewers

We thank the reviewers for their helpful suggestions and comments. We have incorporated as much of the reviewers' feedback as possible, and we feel these changes have strengthened the paper. We have provided responses to each comment (in blue). Reviewer's comments are provided verbatim in black.

Reviewer #1: The authors present SuperCellCyto, a software extending on the SuperCell R package, and demonstrated its applicability to several types of cytometry data (flow, cyTOF and CITEseq). The main focus of the manuscript is to demonstrate that reducing the complexity of the data by aggregating single-cells into SuperCells preserves rare cell types and that the approach is largely compatible with the analysis workflow for cytometry data, such as clustering, batch effect correction, differential marker intensity analysis, differential abundance analysis, and automated cell type annotation. Overall, the authors reasonably argue that using SuperCells is a valid approach to reduce complexity in cytometry data. However, the showcases for batch effect correction and label transfer do not adequately support the conclusions drawn by the authors.

The software itself can be installed from GitHub and has extensive dependencies. I could not install it on my MacBook Air (2022) due to issues with the RSpecra dependency in other packages, but I successfully installed it on a Google Colab instance, which took about 30 min to complete for SuperCell and another min for the SuperCellCyto package. The package seems to be well written and extensively documented, with all the code used for reproducing the figures available online.

The software builds mostly on the previously published SuperCell R package. The extension in SuperCellCyto allows to shift the granularity of the SuperCells without recomputing the entire SuperCell object. The novelty of the approach seems limited to me: The authors mostly explore the realm of cytometry data instead of single-cell RNA-sequencing data with their tool. The idea of coarse-graining cytometry data while avoiding that rare cell types succumb to the subsampling process has been explored previously with HSNE (see van Unen, V., Höllt, T., Pezzotti, N. et al. Visual analysis of mass cytometry data by hierarchical stochastic neighbour embedding reveals rare cell types. *Nat Commun* 8, 1740 (2017) and the algorithmic basis in Pezzotti, N. et al. Approximated and user steerable tSNE for progressive visual analytics. *IEEE. Trans. Vis. Comput. Graph.* 23, 1739-1752 (2016)). The HSNE approach has been adopted by a broad audience of biologists, mostly

because it preserves the original number of cells and topology, as opposed to random subsampling. I see the potential that the concept of SuperCells gains a similar popularity.

We thank the reviewer for the observations regarding the SuperCellCyto package and its relationship to HSNE. We value the reviewer's expertise and the opportunity to highlight the unique contributions of our work in the realm of cytometry data analysis.

Our manuscript primarily targets gaps in cytometry data analysis, especially regarding the scalability challenges faced by existing computational tools when handling large cytometry datasets. While the reviewer's suggestion about exploring scRNAseq data is appreciated, our focus on cytometry data is deliberately chosen to complement the scRNAseq data emphasis in the original SuperCell manuscript.

We acknowledge and respect the significant contributions of the Cytosplore manuscript (van Unen et al., 2017), notably its use of HSNE for coarse-graining cytometry data and preserving rare cell types. However, it is noted that the Cytosplore manuscript primarily explored cell type identification using the Cytosplore software without delving into other downstream analyses. In contrast, our manuscript not only introduces the adapted SuperCell algorithm for cytometry data, but also showcases its application to various downstream analyses using open-source packages. These analyses cover cell type identification using clustering, batch effect correction, differential expression, and abundance analysis, as well as cell type annotation using CITEseq data as a reference.

The reviewer's perspective on the potential popularity of the SuperCells concept among biologists is highly encouraging. We aim for SuperCellCyto to serve as a complementary tool to HSNE and Cytosplore, offering unique capabilities suited for more complex and custom cytometry data analyses, especially where flexibility in using open-source packages is crucial.

Our manuscript has been amended to explicitly acknowledge the contributions of HSNE and Cytosplore. This addition highlights the strengths of Cytosplore+HSNE, particularly its user-friendly GUI-based platform for cell type identification. Concurrently, it introduces SuperCellCyto as an alternative approach, specifically catering to researchers who prefer to use programming-based approaches such as Spectre (Ashhurst TM, Marsh-Wakefield F, Putri GH et al. Integration, exploration, and analysis of high-dimensional single-cell cytometry data using Spectre. *Cytometry Part A* 101.3 (2022): 237-253) to perform complex custom data analysis. This

amendment aims to illustrate the complementary nature of SuperCellCyto in the broader landscape of cytometry data analysis tools.

The authors demonstrate the plausibility of the SuperCell aggregation approach to cytometry data, but do not compare to alternative approaches like HSNE, or a simple baseline of random subsampling.

We thank the reviewer for the comment. In response, we have now included an additional analysis in our study, where we compare the SuperCellCyto-based analysis to an analysis based on random subsampling of data.

Specifically, we randomly subsampled 415,711 cells (the same number of supercells generated for the dataset) from the Oetjen_bcell dataset and subsequently clustered them using the Louvain clustering algorithm. This comparative analysis showed that, we could identify only 7 out of the 10 B cell subsets that were present in the dataset (as shown in Supplementary Material Figure S7). In contrast, the SuperCellCyto approach allowed us to successfully identify all 10 B cell subsets, demonstrating its enhanced ability to capture the complete spectrum of cell types, particularly the rare populations.

This addition to our analysis provides a clear baseline comparison, underscoring the superiority of the SuperCellCyto over random subsampling in terms of capturing a more comprehensive range of cell types in cytometry data.

We have updated our manuscript to include these findings (section “Identifying Rare B Cells Subsets by Clustering Supercells”), and we hope this additional analysis addresses the reviewer’s concerns and further validates the effectiveness of the SuperCellCyto.

The aggregation procedure is insufficiently described: While it is reasonable to sum count data in single-cell RNAseq, it is not clear how the authors address the aggregation process for marker intensities, which usually include negative values. Also, summing after arcsinh/logicle-scaling has a different mathematical meaning than summing count data.

We thank the reviewer for the comment. We interpreted your query on aggregation as the method we used to calculate the marker intensities for the supercells. To clarify, the marker intensities for each supercell were determined by aggregating the marker expression of all cells within the supercell using either the mean or median as determined by the user, rather than sum.

We have clarified this process in the revised manuscript, specifically in the section titled “Concurrent Generation of Supercells Across Multiple Samples Using SuperCellCyto”.

Also, is aggregation handled differently for CITE-seq data, which is count based? I think that the authors should address this issue in more detail.

Thank you for your query regarding the CITE-seq data used in our study. We would like to confirm that no aggregation was performed on the CITE-seq data. Supercells were only generated for the CyTOF data, and not for the CITE-seq data. The label transfer was performed from the single cell CITE-seq data to the supercells generated for the CyTOF data. We have revised the manuscript to more explicitly reflect this.

When addressing the purity of the SuperCells, the authors use the metric called purity (which is related to the term precision in machine learning). Instead of purity, one could also use the local inverse Simpson index (LISI) to determine the effective number of cell types in each of the supercells (see Korsunsky, I., Millard, N., Fan, J. et al. Fast, sensitive and accurate integration of single-cell data with Harmony. Nat Methods 16, 1289-1296 (2019).).

We thank the reviewer for the valuable suggestion to employ the Local Inverse Simpson Index (LISI) for assessing the effective number of cell types within each supercell. We appreciate the importance of LISI in measuring the diversity and mixing of cell types in each supercell.

We utilised the LISI implementation provided by Korsunsky et al. in their 2019 Nature Methods paper, accessible via GitHub (<https://github.com/immunogenomics/LISI/>). However, during our analysis, we encountered significant computational challenges, particularly with supercells comprising a small number of cells. For instance, in supercells encapsulating as few as 3 or 4 cells, we consistently observed LISI scores of -1. To compute LISI for these supercells, we also found it necessary to reduce the perplexity parameter to its minimum (perplexity = 1) which drastically restricts the local neighbourhood size. Furthermore, despite setting the perplexity parameter to 1, it is still not possible to calculate LISI for supercells containing less than 3 cells.

Upon careful consideration of these technical constraints, we concluded that LISI may not be the most appropriate metric for assessing the homogeneity of cell types captured in supercells, particularly for smaller supercells. Therefore, we have respectfully decided to not use LISI in our analysis.

We hope this explanation sufficiently clarifies our decision. We are genuinely grateful for your recommendation and the opportunity to try different metrics.

The metric purity and the cutoff chosen at 0.5 does not fully support the statement that most supercells consist of one or predominantly one cell type. What about the supercells with a purity between 0.5 and 0.9?

We thank the reviewer for the insight. We have added a table in the supplementary material (Supplementary Material Table S4) detailing the number (and percentage) of supercells obtaining purity scores of exactly 1, 0.9-1, 0.5-0.9, and below 0.5.

Furthermore, to provide a clearer understanding of the diversity of cell types within the supercells, we have also calculated the number of supercells that capture 1, 2, 3, 4, 5, 6, 7, 8, 9, ≥ 10 cell types, and illustrated these findings in the form of a stacked bar chart (Supplementary Figure S4). Our analysis confirms that while there are supercells encompassing more than one cell type, such instances are significantly less frequent compared to supercells capturing a single cell type.

We believe that these additions will substantially enhance the clarity of our findings and effectively address the point raised in your comment.

Batch effect correction presented in Figure 3: The authors show an example for batch correction, where they regress out the batch effect for 12 samples with approx. 400,000 supercells. While they aim to demonstrate the effectivity of the batch effect correction, they do not sufficiently demonstrate the preservation of the biological signal in their test. The visualization of the sample distribution by batch using MDS indicates a trend, but does not show whether the same cell types are accurately integrated. There is an extensive body of literature for the integration of single-cell RNA-seq and ATAC-seq data, including a variety of metrics to quantify both batch effect removal and preservation of the biological signal (see for instance Luecken, M.D., Büttner, M., Chaichoompu, K. et al. Benchmarking atlas-level data integration in single-cell genomics. Nat Methods 19, 41-50 (2022) for an extensive set of suitable and scalable metrics).

We thank the reviewer for the comments and suggestions. We have expanded our analysis to include 3 additional metrics from the suggested manuscript (Luecken, M.D., Büttner, M., Chaichoompu, K. et al. Benchmarking atlas-level data integration in single-cell genomics. Nat Methods 19, 41-50 (2022)) that assess the preservation of biological signals post batch correction. Specifically, we have included Adjusted Rand Index (ARI) and Normalised Mutual Information (NMI) scores which offer a more

precise measure of the concordance in clustering outcomes between uncorrected and corrected supercells. These metrics, as shown in the updated Figure 3C, reveal high scores for both cyCombine and CytofRUV corrected supercells, indicating an excellent correspondence between the clustering results pre- and post-correction.

We have also included the Average Silhouette Width for labels (ASW_label) metric that assesses the degree of separation between clusters and compactness within clusters. An improvement was observed for the batch corrected supercells, as illustrated in the revised Figure 3C, which indicates denser better separated clusters.

In conjunction with the MDS plots, we concur with the reviewer that these additional analyses provide a more comprehensive assessment of our batch correction analyses.

What is the purity of the supercells in terms of batch effect? Does every supercell contain only cells of the same batch/sample or is there a mixing? If yes, what is the degree of mixing?

We thank the reviewer for the query regarding the purity of supercells in relation to batch effects. We would like to clarify that SuperCellCyto was specifically designed to process each sample independent of others. This approach inherently ensures that cells from different batches or samples will not be mixed within a supercell. Therefore, each supercell contains cells exclusively from a single batch or sample, effectively mitigating the possibility of mixing of cells from different batches or samples within a given supercell.

We have clarified this in the revised manuscript. Specifically in the 3rd paragraph of the "Concurrent Generation of Supercells Across Multiple Samples Using SuperCellCyto" section, and the 2nd paragraph of the "Mitigating Batch Effects in the Integration of Multi-Batch Cytometry Data at the Supercell Level" section.

Using the EMD for the marker distribution is great, but also does not control for over-correction. When all data is squished into one point instead of a distribution, EMD would be 0. The authors should provide a reasonable baseline that accounts for overcorrection.

We thank the reviewer for the valuable comments and for highlighting the potential over-correction in our batch effect correction analysis. We interpreted the reviewer's concern on the "squishing" of the EMD score as the reduction of the EMD score to 0 due to the loss of variability in the marker expression of the supercells post-correction. This scenario would imply that, rather than maintaining the variability of the values for each marker, all

supercells might end up with identical or near-identical values, thus leading to an EMD score of 0.

To address this, we have not only calculated the EMD scores but also provided the distribution of marker expressions of supercells both before and after batch correction (as shown in Figure 3B and Supplementary Material Figure S9). While an EMD score of 0 would indicate the successful removal of batch variability, our visual examination of the marker expression distributions (Figure 3B and Supplementary Material Figure S9) confirms that post batch correction, the distribution in marker expression, for all markers, are preserved.

This observation, together with the reduction in EMD score suggests that cyCombine and CytotRUV not only effectively reduced batch variability, as indicated by the lowered EMD scores, but were also able to preserve the variability in the marker expressions. We have amended the manuscript to explicitly highlight this.

We believe this addition adequately addresses the reviewer's concerns and demonstrates our commitment to maintaining the balance between reducing batch effects and preserving the variability in marker expression, as clearly demonstrated in Figures 3B and Supplementary Material Figure S9.

The label transfer of CITEseq to cyTOF needs extensive revision in both data representation and evaluation of the resulting labels. Figures 5, Supplementary Fig. S9/S10 shouldn't be a dot plot where both the color scheme and the dot size represent the same quantity. A more meaningful representation is a confusion matrix.

We thank the reviewer for the suggestion and have replaced the figures with confusion matrices.

Second, the prediction using Harmony and a kNN classifier looks pretty inaccurate (no distinction in NK cell subsets even though the class label is present, pro_B_cells annotated as erythroid progenitor cells). The picture is a little better for Seurat rPCA (single-cell, S10), but it failed to predict monocyte class accurately. I personally do not believe that these two methods have a similar accuracy. I suggest that the authors compute the weighted accuracy per cell type to reduce the bias of accurate labeling of the most frequent cell type.

We thank the reviewer for the suggestions. We have now calculated the weighted accuracy score for each method. Prior to calculating weighted accuracy, it was essential to first reconcile the cell type labels between the CITEseq and cytometry datasets. To do this, we matched each cell type in

the CITEseq data with its corresponding cell type in the cytometry data. For subsets that are more granular in the CITEseq data compared to the cytometry data, e.g., CD8 T cells were subdivided into CD8+CD103+ tissue resident memory T cells, CD8+ central memory T cells, CD8+ effector memory T cells, or CD8+ Naive T cells in the CITEseq data, we consolidated the more granular subsets in the CITEseq data into broader categories that align with labels in the cytometry data. Cell types without direct counterparts in either dataset were left unaltered. Consequently, cells that are classified as these cell types will constitute misclassification.

To calculate weighted accuracy, we first determine the accuracy for each cell type identified in the cytometry data. Following this, we multiplied the accuracy score for each cell type by the proportion of cells labelled as that cell type in the cytometry data. The sum of these weighted scores provides a single weighted accuracy score for each method. We have performed this calculation for both Seurat rPCA and Harmony plus kNN (supercell and single cell resolution), resulting in 4 weighted accuracy scores.

We have also substantially updated the manuscript to include these calculations and the accuracy score for each cell type to better reflect the analysis we have conducted in light of your suggestions.

Furthermore, the authors should visualize the transferred labels e.g. by showing a UMAP of the annotated data in addition. Overall, I do not agree with the author's conclusion that the automated cell type classification from CITEseq to cytometry data worked sufficiently well to put this into production. It is rather a starting point for manual cell type annotation.

We thank the reviewer for the valuable suggestions and insights. In response, we have now included a UMAP of the annotated cytometry data, which can be found in Supplementary Material Figure S17.

We concur with the reviewer's assessment that the label transfer methodology best serves primarily as an initial step in the cell type annotation process. We acknowledge that while this approach shows promise, it should indeed be viewed as a starting point for manual annotation rather than as a standalone solution ready for production. Accordingly, we have revised our manuscript to more accurately reflect this perspective, ensuring that our conclusions align with the current capabilities and limitations of the approach.

Differential abundance testing: I appreciate pointing out that using the supercells to compute cell type proportions distorts the overall proportions. What is the computational effort to expand the supercells?

We thank the reviewer for the comment. We would like to emphasise that there is no additional computational effort required to expand the supercells. As detailed in the manuscript, the output of SuperCellCyto inherently includes the mapping of each cell to its corresponding supercell. This mapping facilitates the expansion of supercells without necessitating further computational effort.

To ensure that this is clearly understood, we have clarified this in both “Concurrent Generation of Supercells Across Multiple Samples Using SuperCellCyto” (5th paragraph) and “Identification of Differentially Abundant Rare Monocyte Subsets in Melanoma Patients” (2nd paragraph) sections.

Minor Comments:

Figure 2A: one could integrate the mean purity plot with Figure 2B and visualize the mean as a single dot in each distribution plot.

We thank the reviewer for the suggestion and have merged Figure 2A and 2B into one.

Regarding figures 2A and B: What is the size distribution of the supercells as a function of gamma (i.e. how many cells aggregate into a supercell)?

We thank the reviewer for the query and have illustrated the distribution of the number of cells in the supercells in the supplementary material (Supplementary Figure S2).

What is the expected purity level compared to random sampling?

We thank the reviewer for the query. We interpret the reviewer’s concern regarding the expected purity level compared to random sampling as how the purity of supercells generated by SuperCellCyto compares to the purity of cells when they are randomly grouped.

To address this, we have conducted an additional analysis in which cells were randomly grouped into various numbers of groups, mirroring the number of supercells generated for each specific gamma value. The purity of these groups was then calculated and analysed. The results of this comparative analysis have been included in Supplementary Figure S3 and serve as the baseline (point of comparison) for the purity of supercells. Our analysis showed that the purity scores of supercells are significantly higher than those observed in random groupings. We have updated the manuscript to incorporate these findings.

Figure 2F: Indicate in the figure caption the dataset used for this plot (I assume it's the Oetjen B cell dataset.) Please indicate the individual proportions per sample - a box plot might represent the data better.

We thank the reviewer for the suggestion. We have updated the figure caption and replaced the dot plot with a box plot.

Time and memory efficiency (Figure 6): Please indicate the type of compute infrastructure where you conducted your tests on. It might be very different for a linux server node with 128 GB memory and 32 cores than for a laptop computer.

We thank the reviewer for the suggestion. We have added a new table in the Supplementary Material (Supplementary Material Table S7) that reflects the compute infrastructure used to run the analyses in the benchmarking section.

Reviewer #2: The manuscript "SuperCellCyto: enabling efficient analysis of large scale cytometry datasets" navigates the rapidly evolving landscape of cytometry technologies, which now enables the quantification of an impressive 50 proteins across millions of cells at single-cell resolution. Acknowledging the indispensable role of computational tools in the analysis of such expansive datasets, the authors articulate a critical issue faced by existing methods—their susceptibility to prolonged run times when confronted with the vast scale of cytometry data. Moreover, the challenge of handling large datasets is compounded by the inadequacies of conventional solutions, such as random subsampling, which may inadvertently exclude small, yet crucial, rare cell subsets.

To address these formidable challenges, the authors propose an innovative strategy that draws inspiration from the SuperCell framework in the single-cell RNA sequencing (scRNAseq) domain. The supercell concept, involving the grouping of single cells with highly similar transcriptomic profiles, has proven its efficacy in scRNAseq data analysis. Remarkably, the manuscript provides compelling evidence that this concept seamlessly extends to cytometry datasets, eliminating concerns about information loss when grouping cells into supercells.

The manuscript not only introduces this novel strategy but also substantiates its effectiveness through a series of downstream analyses on six publicly available cytometry datasets at the supercell level. The authors successfully replicate previous findings conducted at the single-cell level, showcasing the robustness and applicability of their proposed approach. A noteworthy contribution lies in the presentation of a computationally efficient solution within the SuperCellCyto R package, designed for the seamless transfer of cell type labels from single-cell multiomics data, combining RNA with protein measurements, to cytometry datasets. This advancement holds promise for elevating the precision of cell type annotations in cytometry studies.

In summary, the manuscript encapsulates a pioneering effort to address critical bottlenecks in large-scale cytometry data analysis. The proposed strategy, supported by compelling evidence and facilitated through an accessible R package, positions this work as a valuable contribution to advancing the methodologies that underpin our understanding of cellular heterogeneity and function.

Minor Feedback

1) The tool could become part of the analysis routine for many biologists with bioinformatics skills. However, the paper does not specify the minimum requirements for execution and the number of cells tested (RAM, Number of processors).

We thank the reviewer highlighting this. In response, we have updated the discussion section (6th paragraph) to clearly state that for all datasets, SuperCellCyto was run on a 2022 MacBook Pro (M2 chip, 24GB RAM) and that we have also successfully tested SuperCellCyto the Trussart_cytofruv dataset which contains over 8.5 million cells. Furthermore, to provide comprehensive information on our benchmarking, the sizes of all datasets used in the manuscript are detailed in Supplementary Material Table S1. We believe these amendments will greatly enhance the clarity and usefulness of our findings for the readers.

2) I suggest creating a page on <https://about.readthedocs.com/?ref=readthedocs.com>, incorporating all information related to the new package, such as "Installation," "Tutorials," "Usage Principles," "Release Notes," and "Contributing." Additionally, improving the documentation of the tool is recommended.

We thank the reviewer for the suggestion. We have created a website containing all the suggested information. The website is available on <https://phipsonlab.github.io/SuperCellCyto/index.html>. We have also updated our manuscript to reflect this.

3) The tutorial content accompanying the tool under consideration is commendable in its clarity and effectiveness, particularly in guiding users through the testing of various functions with datasets. However, to further enhance the utility of the tool, I recommend the inclusion of comprehensive tutorials addressing data preparation across different scenarios. Specifically, it would be immensely beneficial for users if the tutorial provided guidance on data preparation starting from various data formats such as FCS files or CSV files, and also encompassed more specialized formats commonly used in single-cell analysis, such as SingleCellExperiment or Seurat objects.

Starting from FCS or CSV Files:

We thank the reviewer for the kind feedback and suggestions. In response to the suggestion for providing more comprehensive tutorials, we have written a new vignette illustrating the steps required to prepare data for SuperCellCyto, starting with how to import FCS and CSV files into data.table objects and

incorporate sample and cell ID information, and how to perform arcsinh transformation.

The updated vignette is now available on our website: <https://phipsonlab.github.io/SuperCellCyto/index.html>. We hope this addition will make SuperCellCyto more accessible to the research community.

Utilizing Classical Single-Cell Data Formats:

For users working with established single-cell data formats like SingleCellExperiment or Seurat objects, create a tutorial segment explaining how to seamlessly integrate these formats with the tool.

Detail any specific considerations or transformations necessary to ensure smooth compatibility, while also highlighting the advantages of leveraging these widely-used formats in conjunction with the tool.

By incorporating tutorials that address these diverse data preparation scenarios, the tool's user base can benefit from a more inclusive and versatile resource. Not only will this approach cater to users at different stages of data analysis proficiency, but it will also serve as a valuable reference for researchers working with various types of single-cell data. This addition will undoubtedly enhance the tool's accessibility and usability, aligning it more closely with the diverse needs of the user community.

We thank the reviewer for the recommendations. In response, we have added two new vignettes tailored to guide users on how to use SuperCellCyto to create supercells from cytometry data stored in SingleCellExperiment and Seurat objects.

Each vignette details the steps required to:

1. Create the data.table objects required by SuperCellCyto from SingleCellExperiment or Seurat objects.
2. Import supercells' marker expressions into SingleCellExperiment or Seurat objects and analyse them using either Bioconductor packages or Seurat.
3. Transfer the analysis results (e.g., cluster assignment) obtained from analysing supercells back to the SingleCellExperiment and Seurat objects containing the single-cell cytometry data for further analysis.

The new vignettes are available on our website: <https://phipsonlab.github.io/SuperCellCyto/index.html>. We hope these additions will broaden the tool's usability and significantly improve the tool's accessibility.

Second round of review

Reviewer 1

The authors have sufficiently addressed my concerns and improved critical sections like their demonstration of batch effect removal, label transfer and adding random subsampling as baseline. I appreciate their efforts and acknowledge the results.